# RANK THE EPISODES:
# A SIMPLE APPROACH FOR EXPLORATION IN PROCEDURALLY-GENERATED ENVIRONMENTS

**Daochen Zha**[1]**, Wenye Ma**[2]**, Lei Yuan**[2]**, Xia Hu**[1]**, Ji Liu**[2]
[1]Department of Computer Science and Engineering, Texas A&M University
[2]AI Platform, Kwai Inc.
{daochen.zha,xiahu}@tamu.edu
{mawenye,leiyuan03}@kuaishou.com, jiliu@kwai.com

## ABSTRACT

Exploration under sparse reward is a long-standing challenge of model-free reinforcement learning. The state-of-the-art methods address this challenge by introducing intrinsic rewards to encourage exploration in novel states or uncertain environment dynamics. Unfortunately, methods based on intrinsic rewards often fall short in procedurally-generated environments, where a different environment is generated in each episode so that the agent is not likely to visit the same state more than once. Motivated by how humans distinguish good exploration behaviors by looking into the entire episode, we introduce RAPID, a simple yet effective episode-level exploration method for procedurally-generated environments. RAPID regards each episode as a whole and gives an episodic exploration score from both per-episode and long-term views. Those highly scored episodes are treated as good exploration behaviors and are stored in a small ranking buffer. The agent then imitates the episodes in the buffer to reproduce the past good exploration behaviors. We demonstrate our method on several procedurally-generated MiniGrid environments, a first-person-view 3D Maze navigation task from MiniWorld, and several sparse MuJoCo tasks. The results show that RAPID significantly outperforms the state-of-the-art intrinsic reward strategies in terms of sample efficiency and final performance. The code is available at https://github.com/daochenzha/rapid.

## 1 INTRODUCTION

Deep reinforcement learning (RL) is widely applied in various domains (Mnih et al., 2015; Silver et al., 2016; Mnih et al., 2016; Lillicrap et al., 2015; Andrychowicz et al., 2017; Zha et al., 2019a; Liu et al., 2020). However, RL algorithms often require tremendous number of samples to achieve reasonable performance (Hessel et al., 2018). This sample efficiency issue becomes more pronounced in sparse reward environments, where the agent may take an extremely long time before bumping into a reward signal (Riedmiller et al., 2018). Thus, how to efficiently explore the environment under sparse reward remains an open challenge (Osband et al., 2019).

To address the above challenge, many exploration methods have been investigated and demonstrated to be effective on hard-exploration environments. One of the most popular techniques is to use intrinsic rewards to encourage exploration (Schmidhuber, 1991; Oudeyer & Kaplan, 2009; Guo et al., 2016; Zheng et al., 2018; Du et al., 2019). The key idea is to give intrinsic bonus based on uncertainty, e.g., assigning higher rewards to novel states (Ostrovski et al., 2017), or using the prediction error of a dynamic model as the intrinsic reward (Pathak et al., 2017). While many intrinsic reward methods have demonstrated superior performance on hard-exploration environments, such as Montezuma's Revenge (Burda et al., 2018b) and Pitfall! (Badia et al., 2019), most of the previous studies use the same singleton environment for training and testing, i.e., the agent aims to solve the same environment in each episode. However, recent studies show that the agent trained in this way is susceptible to overfitting and may fail to generalize to even a slightly different environment (Rajeswaran et al., 2017; Zhang et al., 2018a). To deal with this issue, a few procedurally-generated environments

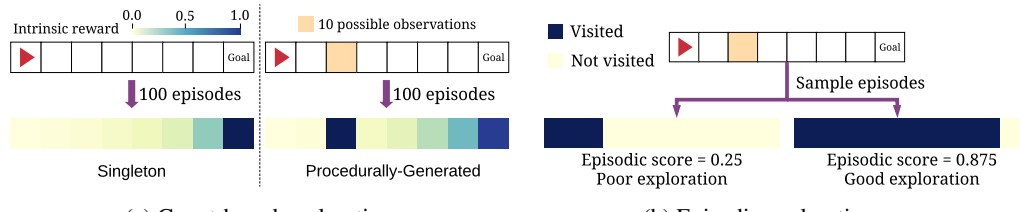

(a) Count-based exploration                  (b) Episodic exploration score

Figure 1: A motivating example of count-based exploration versus episode-level exploration score. While count-based exploration works well in singleton setting, i.e., the environment is the same in each episode, it may be brittle in procedurally-generated setting. In (a), if the observation of the third block in an episode is sampled from ten possible values, the third block will have very high intrinsic reward because the agent is uncertain on the new states. This will reward the agent for exploring the third block and also its neighbors, and hence the agent may get stuck around the third block. In (b), we score behaviors in episode-level with *# visited states/ # total states*. The episodic exploration score can effectively distinguish good exploration behaviors in procedurally-generated setting.

are designed to test the generalization of RL, such as (Beattie et al., 2016; Chevalier-Boisvert et al., 2018; Nichol et al., 2018; Côté et al., 2018; Cobbe et al., 2019; Küttler et al., 2020), in which the agent aims to solve the same task, but a different environment is generated in each episode.

Unfortunately, encouraging exploration in procedurally-generated environments is a very challenging task. Many intrinsic reward methods, such as count-based exploration and curiosity-driven exploration, often fall short in procedurally-generated environments (Raileanu & Rocktäschel, 2020; Campero et al., 2020). Figure 1a shows a motivating example of why count-based exploration is less effective in procedurally-generated environments. We consider a 1D grid-world, where the agent (red) needs to explore the environment to reach the goal within 7 timesteps, that is, the agent needs to move right in all the steps. While count-based exploration assigns reasonable intrinsic rewards in singleton setting, it may generate misleading intrinsic rewards in procedurally-generated setting because visiting a novel state does not necessarily mean a good exploration behavior. This issue becomes more pronounced in more challenging procedurally-generated environments, where the agent is not likely to visit the same state more than once. To tackle the above challenge, this work studies how to reward good exploration behaviors that can generalize to different environments.

When a human judges how well an agent explores the environment, she often views the agent from the episode-level instead of the state-level. For instance, one can easily tell whether an agent has explored a maze well by looking into the coverage rate of the current episode, even if the current environment is different from the previous ones. In Figure 1b, we can confidently tell that the episode in the right-hand-side is a good exploration behavior because the agent achieves $0.875$ coverage rate. Similarly, we can safely conclude that the episode in the left-hand-side does not explore the environment well due to its low coverage rate. Motivated by this, we hypothesize that episode-level exploration score could be a more general criterion to distinguish good exploration behaviors than state-level intrinsic rewards in procedurally-generated setting.

To verify our hypothesis, we propose exploration via **Ra**nking the E**pi**so**d**es (RAPID). Specifically, we identify good exploration behaviors by assigning episodic exploration scores to past episodes. To efficiently learn the past good exploration behaviors, we use a ranking buffer to store the highly-scored episodes. The agent then learns to reproduce the past good exploration behaviors with imitation learning. We demonstrate that RAPID significantly outperforms the state-of-the-art intrinsic reward methods on several procedurally-generated benchmarks. Moreover, we present extensive ablations and analysis for RAPID, showing that RAPID can well explore the environment even without extrinsic rewards and could be generally applied in tasks with continuous state/action spaces.

## 2   Exploration via Ranking the Episodes

An overview of RAPID is shown in Figure 2. The key idea is to assign episodic exploration scores to past episodes and store those highly-scored episodes into a small buffer. The agent then treats the episodes in the buffer as demonstrations and learns to reproduce these episodes with imitation learning. RAPID encourages the agent to explore the environment by reproducing the past good

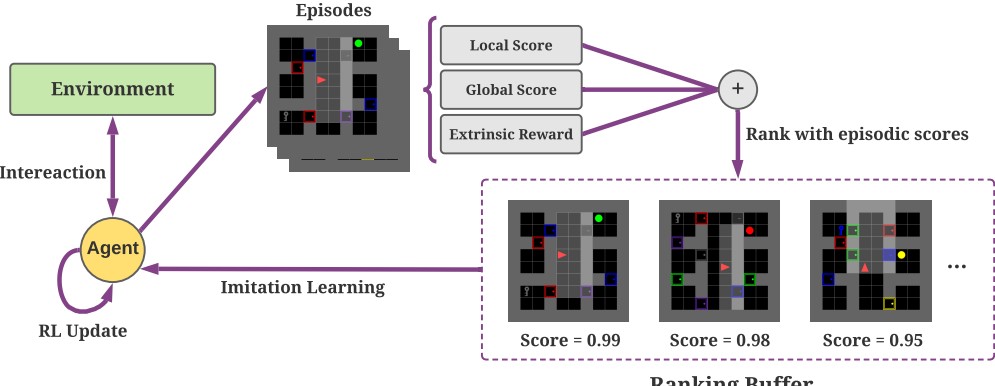

Figure 2: An overview of RAPID. The past episodes are assigned episodic exploration scores based on the local view, the global view, and the extrinsic reward. Those highly scored episodes are stored in a small ranking buffer. The agent is then encouraged to reproduce the past good exploration behaviors, i.e., the episodes in the buffer, with imitation learning.

exploration behaviors. This section introduces how we define the episodic exploration score for procedurally-generated environments and how the proposed method can be combined with state-of-the-art reinforcement learning agents.

## 2.1 EPISODIC EXPLORATION SCORE

Each episode is scored from both local and global perspectives. The local score is a per-episode view of the exploration behavior, which measures how well the current episode itself explores the environment. The global score provides a long-term and historical view of exploration, i.e., whether the current episode has explored the regions that are not well explored in the past. Additionally, we consider extrinsic reward, which is the episodic reward received from the environment. It is an external criterion of exploration since a high extrinsic reward often suggests good exploration in sparse environments. The overall episodic exploration score is obtained by the weighted sum of the above three scores. We expect that these scores can model different aspects of exploration behaviors and they are complementary in procedurally-generated environments.

**Local Score.** The intuition of the local exploration bonus is that the agent is expected to visit as many distinct states as possible in one episode. In our preliminary experiments on MiniGrid, we observe that many poor exploration episodes repeatedly visit the same state. As a result, the agent may get stuck in a well-known region but never reach out to unexplored regions. From a local view, we can quantify this exploration behavior based on the diversity of the visited states. Specifically, we define the local exploration score as

$$S_{\text{local}} = \frac{N_{\text{distinct}}}{N_{\text{total}}},\tag{1}$$

where $N_{\text{total}}$ is the total number of states in the episode, and $N_{\text{distinct}}$ is the number of distinct states in the episode. Intuitively, optimizing the local score will encourage the agent to reach out to the unexplored regions in the current episode. In an ideal case, the episodes that never visit the same state twice will receive a local bonus of $1$. There could be various ways to extend the above definition to continuous state space. In this work, we empirically define the local score in continuous state space as the mean standard deviation of all the states in the episode:

$$S_{\text{local}} = \frac{\sum_{i=1}^{l} \text{std}(s_i)}{l},\tag{2}$$

where $l$ is the dimension of the state, and $\text{std}(s_i)$ is the standard deviation along the $i$-dimension of the states in the episode. Intuitively, this score encourages the episodes that visit diverse states.

**Global Score.** The global score is designed to model long-term exploration behavior. While a different environment will be generated in each episode in the procedurally-generated setting, the environments in different episodes usually share some common characteristics. For example, in the

---

**Algorithm 1** Exploration via Ranking the Episodes (RAPID)

---

1: **Input:** Training steps $S$, buffer size $D$, RL rollout steps $T$
2: Initialize the policy $\pi_\theta$, replay buffer $\mathcal{D}$
3: **for** iteration = 1, 2, ... until convergence **do**
4:     Execute $\pi_\theta$ for $T$ timesteps
5:     Update $\pi_\theta$ with RL objective (also update value functions if any)
6:     **for** each generated episode $\tau$ **do**
7:         Compute episodic exploration score $S_\tau$ based on Eq. (4)
8:         Give score $S_\tau$ to all the state-action pairs in $\tau$ and store them to the buffer
9:         Rank the state-action pairs in $\mathcal{D}$ based on their exploration scores
10:       **if** $\mathcal{D}.length > D$ **then**
11:           Discard the state-action pairs with low scores so that $\mathcal{D}.length = D$
12:       **end if**
13:       **for** step = 1, 2, .., $S$ **do**
14:           Sample a batch from $\mathcal{D}$ and train $\pi_\theta$ using the data in $\mathcal{D}$ with behavior cloning
15:       **end for**
16:     **end for**
17: **end for**

---

MiniGrid MultiRoom environment (Figure 3a), although a different maze will be generated in a new episode, the basic building blocks (e.g., walls, doors, etc.), the number of rooms, and the size of each room will be similar. From a global view, the agent should explore the regions that are not well explored in the past. Motivated by count-based exploration, we define the global score as

$$S_{\text{global}} = \frac{1}{N_{\text{total}}} \sum_s \frac{1}{\sqrt{N(s)}}, \tag{3}$$

where $N(s)$ is the state count of $s$ throughout the training process, that is, the global score is the mean count-based intrinsic reward of all the states in the episode.

**Episodic Exploration Score.** Suppose $S_{\text{ext}}$ is the total extrinsic reward of an episode received from the environment . The episodic exploration score is obtained by the weighted sum of the extrinsic reward $S_{\text{ext}}$, the local score $S_{\text{local}}$, and the global score $S_{\text{global}}$:

$$S = w_0 S_{\text{ext}} + w_1 S_{\text{local}} + w_2 S_{\text{global}}, \tag{4}$$

where $w_0$, $w_1$ and $w_2$ are hyperparameters. We speculate that the three scores are complementary and the optimal $w_0$, $w_1$ and $w_2$ could vary in different environments. For example, a higher $w_1$ will give a higher weight to the per-episode view of exploration behavior. A higher $w_2$ will focus more on whether the current episode has explored the past unexplored regions. Nevertheless, we find that a single set of $w_0$, $w_1$ and $w_2$ works well across many tasks.

**Comparison with Prior Work.** Several papers have studied episode-level feedback to improve the sample efficiency of RL. To the best of our knowledge, the existing work mainly focuses on learning a meta-policy based on episodic rewards (Xu et al., 2018; Zha et al., 2019b), or imitating the episodes with high episodic rewards (Guo et al., 2018; Srivastava et al., 2019). However, in very sparse environments, it is not likely to receive a non-zero reward without exploration. These methods are not suitable for very sparse environments since episodic rewards could be always zero. In this work, we propose local and global episodic scores to quantify the exploration behaviors and encourage the agent to visit distinct states from per-episode and long-term views.

## 2.2 Reproducing Past Good Exploration Behaviors with Imitation Learning

A straightforward idea is to treat the episodic exploration score as a reward signal and assign it to the last timestamp of the episode. In this way, the agent can be updated with any reinforcement learning objective. However, this strategy is not very effective with three potential limitations. First, the reward remains sparse so that the reinforcement learning agents may not be able to effectively exploit the reward signals. Second, the intrinsic exploration bonus may introduce bias in the objective. While the local score and global score may encourage exploration at the beginning stage, they are

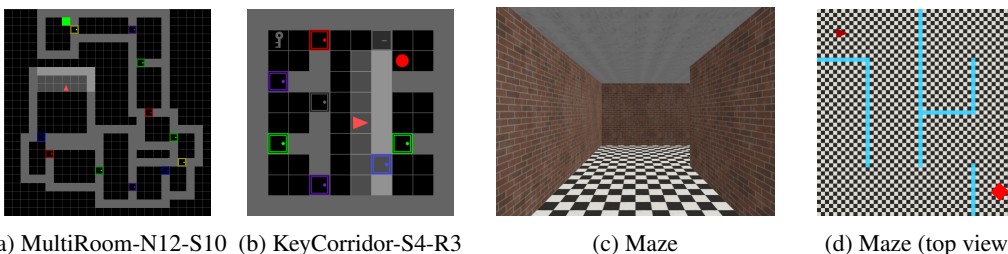

(a) MultiRoom-N12-S10  (b) KeyCorridor-S4-R3  (c) Maze  (d) Maze (top view)

Figure 3: Rendering of the procedually-generated environments in our experiments.

not true objectives and may degrade the final performance. Third, it only uses a good episode once. In fact, many past good episodes can be exploited many times to improve sample efficiency.

To overcome the above limitations, we propose to use a small ranking buffer to store past good state-action pairs and then use imitation objective to enhance exploration. Specifically, we assign the same episodic exploration score to all the state-action pairs in an episode and rank all the state-action pairs based on scores. In addition to the original RL objective, we employ behavior cloning, the simplest form of imitation learning, to encourage the agent to reproduce the state-action pairs in the buffer. This simple ranking buffer can effectively exploit the past good experiences since a good state-action pair could be sampled more than once. The procedure is summarized in Algorithm 1. A possible variant is to keep the entire episodes in the buffer. We have tried this idea and do not observe a clear difference between this variant and Algorithm 1 (see Appendix H for more discussions).

## 3 EXPERIMENTS

The experiments are designed to answer the following research questions: **RQ1:** how is RAPID compared with the state-of-the-art intrinsic reward methods on benchmark procedurally-generated environments (Section 3.2)? **RQ2:** how will RAPID perform if removing one of the scores (i.e., the local score, the global score, and the extrinsic reward) or the buffer (Section 3.3)? **RQ3:** how will the hyperparameters impact the performance of RAPID (Section 3.3)? **RQ4:** can RAPID explore the environment without extrinsic reward (Section 3.3)? **RQ5:** how will RAPID perform with larger and more challenging grid-world (Section 3.3)? **RQ6:** can RAPID generalize to 3D navigation task and continuous state/action space, such as MuJoCo (Section 3.4)?

### 3.1 ENVIRONMENTS AND BASELINES

Figure 3 shows the rendering of procedurally-generated environments used in this work. Following the previous studies of exploration in procedurally-generated environments (Raileanu & Rocktäschel, 2020; Campero et al., 2020), we mainly focus on MiniGrid environments (Chevalier-Boisvert et al., 2018), which provide procedurally-generated grid-worlds. We consider two types of hard exploration tasks: *MultiRoom-NX-SY*, where X and Y indicate the number of rooms and the room size, respectively, and *KeyCorridor-SX-RY*, where X and Y indicate the room size and the number of rows, respectively. A large X or Y suggests a larger environment, which will be more difficult to explore. The agent has a partial view of the environment and needs to navigate the green block or the red ball under sparse rewards. These tasks are challenging because the agent needs to explore many rooms in order to receive a reward signal so that standard RL algorithms will typically fail. To test whether RAPID can generalize to continuous state space, we consider MiniWorld Maze (Chevalier-Boisvert, 2018), whose observation space is $60 \times 80$ images. Similarly, the agent in MiniWorld Maze has a 3D partial view of the environment and needs to navigate the red box. The maze is procedurally-generated, i.e., a different maze will be generated in each episode. We further study RAPID in continuous state and action spaces on sparse MuJoCo tasks (Todorov et al., 2012; Brockman et al., 2016). More details of the above environments are described in Appendx B.

While exploration has been extensively studied in the literature, very few methods are designed for procedurally-generated environments. To better understand the performance of RAPID, we consider the baselines in the following categories. **Traditional Methods:** we consider several popular methods designed for singleton environments, including count-based exploration (COUNT) (Bellemare et al., 2016), Random Network Distillation Exploration (RANDOM) (Burda et al., 2018b),

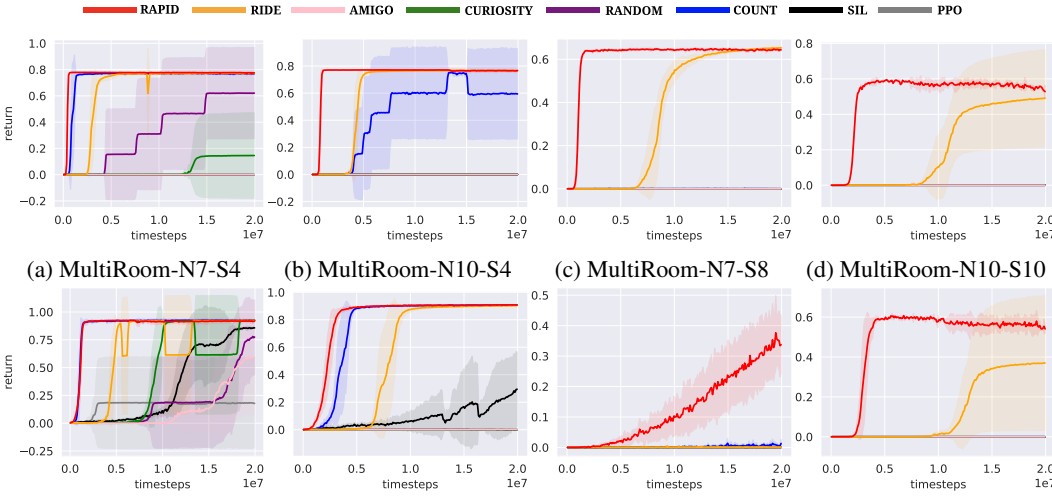

Figure 4: Performance of RAPID against baselines on hard-exploration environments in MiniGrid. All the experiments are run 5 times. The shaded area represents mean ± standard deviation.

|          | RAPID           | w/o local       | w/o global      | w/o reward        | w/o buffer        | w/o ranking       |
|----------|-----------------|-----------------|-----------------|-------------------|-------------------|-------------------|
| MR-N7S4  | **0.787 ± 0.001** | **0.787 ± 0.000** | **0.787 ± 0.001** | 0.781 ± 0.002     | 0.000 ± 0.000     | 0.002 ± 0.002     |
| MR-N10S4 | **0.778 ± 0.000** | **0.778 ± 0.000** | **0.778 ± 0.001** | 0.775 ± 0.002     | 0.000 ± 0.000     | 0.000 ± 0.000     |
| MR-N7S8  | **0.678 ± 0.001** | 0.677 ± 0.002   | 0.677 ± 0.002   | 0.652 ± 0.004     | 0.000 ± 0.000     | 0.000 ± 0.000     |
| MR-N10S10| **0.632 ± 0.001** | 0.238 ± 0.288   | 0.630 ± 0.005   | 0.604 ± 0.010     | 0.000 ± 0.000     | 0.000 ± 0.000     |
| MR-N12S10| **0.644 ± 0.001** | 0.001 ± 0.001   | 0.633 ± 0.005   | 0.613 ± 0.007     | 0.000 ± 0.000     | 0.000 ± 0.000     |
| KC-S3R2  | **0.934 ± 0.004** | 0.933 ± 0.002   | **0.934 ± 0.000** | 0.929 ± 0.003     | 0.018 ± 0.008     | 0.527 ± 0.380     |
| KC-S3R3  | **0.922 ± 0.001** | 0.885 ± 0.022   | 0.912 ± 0.003   | 0.903 ± 0.002     | 0.012 ± 0.007     | 0.013 ± 0.006     |
| KC-S4R3  | **0.473 ± 0.087** | 0.150 ± 0.095   | 0.244 ± 0.144   | 0.035 ± 0.035     | 0.000 ± 0.000     | 0.001 ± 0.001     |

Table 1: Performance of RAPID and the ablations on MiniGrid environments. The mean maximum returns and standard deviations are reported. MR stands for MultiRoom; KC stands for KeyCorridor. Learning curves are in Appendix C.

and curiosity-driven exploration (CURIOSITY) (Pathak et al., 2017). **Methods for Procedurally-Generated Environments:** we consider two state-of-the-art exploration methods, including Impact-Driven Exploration (RIDE) (Raileanu & Rocktäschel, 2020), and exploration with Adversarially Motivated Intrinsic Goals (AMIGO) (Campero et al., 2020). **Self-Imitation Methods**: self-imitation learning (SIL) (Oh et al., 2018) also exploits past good experiences to encourage exploration. We further include PPO (Schulman et al., 2017) as a reinforcement learning baseline. All the algorithms are run the same number timesteps for a fair comparison. More details are provided in Appendix A.

## 3.2 MINIGRID RESULTS

To study **RQ1**, we plot the learning curves of RAPID and the baselines on MiniGrid benchmarks in Figure 4. We can make three observations. First, RAPID performs significantly better than the state-of-the-art methods designed for singleton or procedurally-generated environments in terms of sample efficiency and final performance. For example, on MultiRoom-N7-S8, RAPID is at least 10 times more sample efficient than RIDE. On KeyCorridor-S4-R3, RAPID is the only algorithm that successfully navigates the goal within 20 million timesteps. Second, the traditional intrinsic reward methods perform well on easy procedurally-generated environments but perform poorly on hard-exploration tasks. For example, COUNT performs as well as RAPID on KeyCorridor-S3-R2 but fails on more challenging tasks, such as MultiRoom-N7-S8. This suggests that traditional methods may fall short in challenging procedurally-generated environments, which is consistent with the results in previous work (Raileanu & Rocktäschel, 2020). Third, RAPID outperforms SIL across all the tasks. Recall that both RAPID and SIL learn to reproduce past good experiences. While SIL reproduces experiences with high rewards, RAPID additionally considers local and global exploration scores. This verifies the effectiveness of the episodic exploration scores.

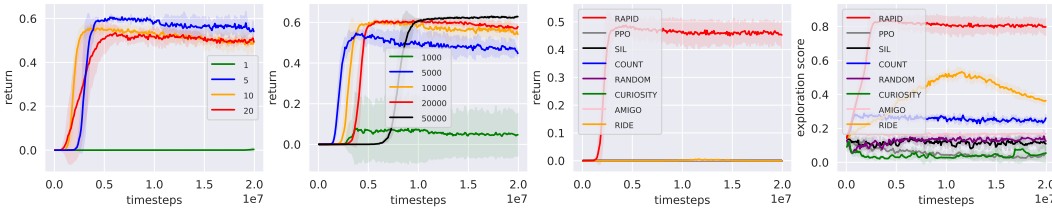

(a) Impact of training steps    (b) Imapct of buffer size    (c) Pure exploration    (d) Local exploration score

Figure 5: Analysis of RAPID on MultiRoom-N12-S10. Results for other environments are in Appendix D. **(a)(b):** the impact of the hyperparameters. **(c)(d):** the returns without extrinsic rewards (pure exploration) and the corresponding local exploration scores.

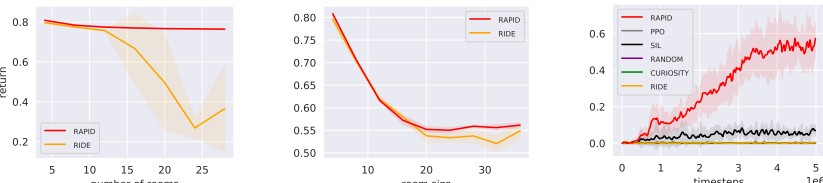

Figure 6: **Left:** the maximum returns w.r.t. the number of rooms under room size 4 (full curves in Appendix E). **Middle:** the maximum returns w.r.t. the room sizes with 4 rooms (full curves in Appendix F). **Right:** learning curves on MiniWolrd Maze (more results in Appendix G).

## 3.3 ABLATIONS AND ANALYSIS

To study **RQ2**, we perform ablation studies on RAPID. Specifically, we remove one of the local score, the global score, and the extrinsic reward, and compare each ablation with RAPID. Besides, we consider a variant that directly uses the episodic exploration score as the reward signal without buffer. To demonstrate the effectiveness of the ranking, we further include a baseline that uses the replay buffer without ranking. Table 1 shows the results. We observe that the local score contributes the most to the performance. While most existing exploration methods reward exploration globally, the global score could be not aware of generalization, which may explain the unsatisfactory performance of COUNT and CURIOSITY. While the global score seems to play a relatively small role as compared to local score in most MiniGrid environments, it plays an important role in KeyCorridor-S4-R3. A possible reason is that the local score is a too weak signal in KeyCorridor-S4-R3, and the global score may help alleviate this issue since it provides some historical information. We also observe that the agent fails in almost all the tasks without buffer or ranking. This suggests that the ranking buffer is an effective way to exploit past good exploration behaviors.

To study **RQ3**, we plot the learning curves with different training steps $S$ and buffer size $D$ in Figure 5a and 5b. We observe that larger number of training steps usually leads to faster learning speed at the beginning stage but may result in sub-optimal performance. For the buffer size, we observe that a large buffer will lead to slow learning speed with more stable performance. We speculate that there is a trade-off between learning speed and the final performance. In our experiments, we set training steps $S$ to be 5 and buffer size $D$ to be 10000, respectively, across all the tasks.

To investigate **RQ4**, we remove extrinsic rewards and show the learning curves of RAPID and the baselines in Figure 5c with the corresponding local exploration scores in Figure 5d. We make two interesting observations. First, without extrinsic rewards, RAPID can deliver reasonable performance. For example, on MultiRoom-N12-S10, RAPID achieves nearly 0.5 average return with pure exploration, compared with around 0.6 if using extrinsic rewards. We look into the episodes generated by RAPID with pure exploration and observe that, in most cases, the agent is able to reach the goal but struggles to find the optimal path to the goal. This is expected because the extrinsic reward is obtained by whether the agent can reach the goal subtracting some penalties of the timesteps spent. The extrinsic reward can encourage the agent to choose the shortest path to reach the goal. For other baselines, we observe that only RIDE and AMIGO can occasionally reach the goal. Second, we observe that some baselines, such as RIDE, are indirectly maximizing the local exploration scores. For example, we observe that the local exploration score increases for RIDE on MultiRoom-N12-S10. However, the local score of RAPID drops after about 10 million timesteps. This suggests the episode-level score may better align with good exploration behaviors. A possible explanation for the superiority of RAPID is that it is designed to directly optimize episodic exploration score.

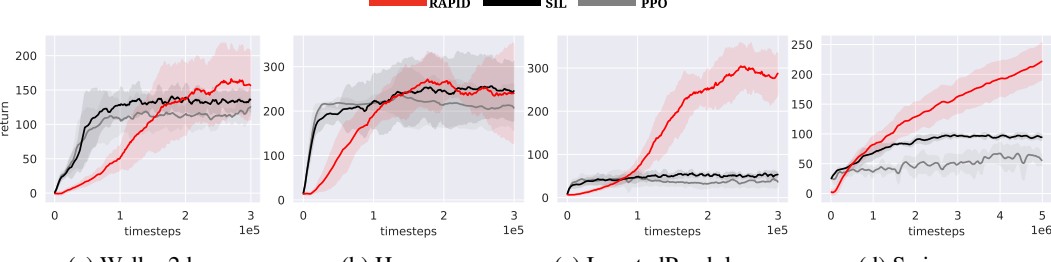

| (a) Walker2d | (b) Hopper | (c) InvertedPendulum | (d) Swimmer |

Figure 7: Performance on MuJoCo tasks with episodic reward. i.e., a non-zero reward is only provided at the end of an episode. All the experiments are run 5 times with different random seeds.

To answer **RQ5**, we focus on *MultiRoom-NX-SY* and ambitiously vary $X$ and $Y$ to make the environments more challenging. Specifically, we fix $Y$ to be 4 and vary $X$ in the left-hand-side of Figure 6, and fix $X$ to be 4 and vary $Y$ in the middle of Figure 6. Compared with RIDE, our RAPID delivers stronger performance on more challenging mini-worlds.

### 3.4 MINIWORLD MAZE AND SPARSE MUJOCO RESULTS

For **RQ6**, the learning curves on 3D Maze are shown in the right-hand-side of Figure 6. RAPID performs significantly better than the baselines. We also conduct experiments on a subset of sparse MuJoCo tasks (Figure 7). The results show that RAPID improves over PPO and SIL. Interestingly, RAPID achieves more than 200 average return on Swimmer, which surpasses some previous results reported on the dense reward counterpart (Duan et al., 2016; Henderson et al., 2018; Lai et al., 2020). Note that the MuJoCo environments are singleton. We use these environments solely to test whether RAPID can deal with continuous state/action spaces. We have also tried adapting Swimmer to be procedurally-generated and observe similar results (Appendix I)

## 4 RELATED WORK

**Count-Based and Curiosity-Driven Exploration.** These two classes of intrinsic motivations are the most popular and proven to be effective in the literature. Count-based exploration encourages the agent to visit the novel states, which is first studied in tabular setting (Strehl & Littman, 2008) and has been recently extended to more complex state space (Bellemare et al., 2016; Ostrovski et al., 2017; Tang et al., 2017; Martin et al., 2017; Machado et al., 2018; Osband et al., 2019). Curiosity-driven exploration learns the environment dynamics to encourage exploration (Stadie et al., 2015; Pathak et al., 2017; Sukhbaatar et al., 2018; Burda et al., 2018a). However, these methods are designed for singleton environments and often struggle to generalize to procedurally-generated environments (Raileanu & Rocktäschel, 2020).

**Exploration for Procedurally-Generated Environments.** Several recent studies have discussed the generalization of reinforcement learning (Rajeswaran et al., 2017; Zhang et al., 2018a;b; Choi et al., 2018) and designed procedurally-generated environments to test the generalization of reinforcement learning (Beattie et al., 2016; Nichol et al., 2018; Küttler et al., 2020). More recent papers show that traditional exploration methods fall short in procedurally-generated environments and address this issue with new exploration methods (Raileanu & Rocktäschel, 2020; Campero et al., 2020). This work studies a new perspective of exploration bonus in episode-level and achieves significantly better performance than the previous methods on procedurally-generated benchmarks.

**Experience Replay Buffer and Self-Imitation Learning.** Experience replay mechanism is widely used and is initially designed to stabilize deep reinforcement learning (Lin, 1992; 1993; Mnih et al., 2015; Zhang & Sutton, 2017; Zha et al., 2019b; Fedus et al., 2020). Self-Imitation Learning extends the experience replay buffer and uses it to store past good experiences (Oh et al., 2018; Guo et al., 2018; 2019; Gangwani et al., 2018). However, Self-Imitation only exploits highly rewarded experiences and thus is not suitable for very sparse environments. Our work incorporates local and global scores to encourage past good exploration behaviors in sparse procedurally-generated environments.

**Episodic Memory.** Inspired by the episodic memory of the animals, some studies propose to use episodic memory to enhance sample efficiency (Blundell et al., 2016; Pritzel et al., 2017). The key idea is to memorize and replay the good episodic experiences. More recently, episodic mem-

ory is used to generate intrinsic rewards to guide exploration (Savinov et al., 2018). Never Give Up (NGU) (Badia et al., 2019) further considers local and global novelties in the intrinsic rewards. Our work differs from NGU in that we approach the sample efficiency from a different angle by treating each episode as a whole and using a ranking mechanism to select good experiences. Thus, good experiences can be exploited multiple times to improve sample efficiency.

## 5 LIMITATIONS AND DISCUSSIONS

RAPID is designed to study our hypothesis that imitating episode-level good exploration behaviors can encourage exploration in procedurally-generated environments. In the presented algorithm, we have made some simple choices. In order to make RAPID more competitive to the state-of-the-art methods in more complicated environments, in this section, we highlight some limitations of RAPID and the potential research directions.

First, in our work, the local and global scores are mainly calculated on low-dimensional state space. We have not yet tested the scores on more complex procedurally-generated environments, such as ProcGen benchmark (Cobbe et al., 2019), TextWorld (Côté et al., 2018), and NetHack Learning Environment (Küttler et al., 2020), some of which require high-dimensional image inputs. In this context, count-based scores may not be well-suited. Learning the state embedding (Raileanu & Rocktäschel, 2020) and pseudo-counts (Bellemare et al., 2016; Ostrovski et al., 2017) could be the potential ways to address this issue. Moreover, high extrinsic rewards may not necessarily mean good exploration behaviors in these environments. The applicability of RAPID in these environments needs future research efforts.

Second, the ranking buffer greedily selects the highly-reward state-action pairs, which is the key to boosting the performance. However, with this greedy strategy, some highly-ranked state-action pairs may never be forgotten even if they are out-of-date, which may introduce bias. As shown in Figure 5b, a larger buffer size will lead to more stable performance but will sacrifice the sample efficiency. In more complicated environments, selecting the proper buffer size could be non-trivial. Some forgetting mechanism (Novati & Koumoutsakos, 2019) and meta-learning strategies (Zha et al., 2019b) could be the potential directions to better manage the buffer.

Third, in our implementation, we employ behavior cloning, the simplest form of imitation learning, to train the policy. In recent years, many more advanced imitation learning methods have been proposed (Ho & Ermon, 2016; Hester et al., 2017). In particular, the idea of inverse reinforcement learning (Abbeel & Ng, 2004) is to learn reward functions based on the demonstrations. An interesting future direction is to study whether we can derive intrinsic rewards based on the experiences in the buffer via inverse reinforcement learning. Specifically, is there a connection between RAPID and intrinsic rewards from the view of inverse reinforcement learning? Can RAPID be combined with intrinsic rewards to improve the performance? We hope that RAPID can facilitate future research to understand these questions.

## 6 CONCLUSIONS AND FUTURE WORK

This work studies how we can reward good exploration behaviors in procedurally-generated environments. We hypothesize that episode-level exploration score could be a more general criterion to distinguish good exploration behaviors. To verify our hypothesis, we design RAPID, a practical algorithm that learns to reproduce past good exploration behaviors with a ranking buffer. Experiments on benchmark hard-exploration procedurally-generated environments show that RAPID significantly outperforms state-of-the-art intrinsic reward methods in terms of sample efficiency and final performance. Further analysis shows that RAPID has robust performance across different configurations, even without extrinsic rewards. Moreover, experiments on a 3D navigation task and a subset of sparse MuJoCo tasks show that RAPID could generalize to continuous state/action spaces, showing the promise in rewarding good episode-level exploration behaviors to encourage exploration. In the future, we will extend our idea and test it in more challenging procedurally-generated environments. We will also explore the possibility of combining the episode-level exploration bonus with intrinsic rewards. Last, we will try other imitation learning techniques beyond behavior cloning.

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

## A HYPERPARAMETERS AND NEURAL NETWORK ARCHITECTURE

For the proposed RAPID, the common hyperparameters are summarized in Table 2. RAPID is based on the PPO implementation from OpenAI baselines[1]. The nstep is 128 for all MiniGrid environments and MuJoCo environments, and 512 for MiniWorld-Maze-S5. For MuJoCo environments, we report the results with only imitation loss since we find that the policy loss of PPO will harm the performance in the episodic reward setting. The learning rate is $10^{-4}$ for all MiniGrid environments and MiniWorld-Maze-S5, and $5 \times 10^{-4}$ for MuJoCo environments. Since MiniWorld-Maze-S5 and MuJoCo environments have continuous state space, we set $w_2 = 0$. In practice, in MiniWorld-Maze-S5, we observe that counting the states will usually lead to memory error because it is not likely to encounter the same state again in MiniWorld-Maze-S5 (COUNT is excluded in comparison due to memory issues). Note that it is possible to use pseudo-counts (Bellemare et al., 2016; Ostrovski et al., 2017), which we will study in our future work. For MiniGrid-KeyCorridorS2R3-v0, MiniGrid-KeyCorridorS3R3-v0 and MiniWorld-Maze-S5, the update frequency of imitation learning is linearly annealed to 0 throughout the training process. For other environments, the imitation learning is performed after each episode. We keep all other hyperparameters of PPO as default and use the default CNN architecture provided in OpenAI baselines for MiniWorld-Maze-S5 and MLP with 64-64 for other environments. The PPO also uses the same hyperparameters. We summarize the state space of each environment and how the local and global scores are computed in Table 3.

| Hyperparameter | Value |
|---|---|
| $w_0$ | 1 |
| $w_1$ | 0.1 |
| $w_2$ | 0.001 |
| buffer Size | 10000 |
| batch Size | 256 |
| number of update steps | 5 |
| entropy coefficient | 0.01 |
| value function coefficient | 0.5 |
| $\gamma$ | 0.99 |
| $\lambda$ | 0.95 |
| clip range | 0.2 |

Table 2: Common hyperparameters of the proposed RAPID (top) and the hyperparameters of PPO baseline (bottom).

| Environment | State Space | Local Score | Global Score |
|---|---|---|---|
| 8 MiniGrid Environments | $7 \times 7 \times 3$, discrete | Eq. 1 | Eq. 3 |
| MiniWorld Maze | $60 \times 80 \times 3$, continuous | Eq. 2 | No |
| MuJoCo Walker2d | 17, continuous | Eq. 2 | No |
| MuJoCo Walker2d | 11, continuous | Eq. 2 | No |
| MuJoCo Walker2d | 4, continuous | Eq. 2 | No |
| MuJoCo Walker2d | 8, continuous | Eq. 2 | No |

Table 3: The calculation of local and global scores for all the environments. Note that if the state space is continuous. We disable the global score since counting continuous states is meaningless. A possible solution is to use pseudo-counts (Bellemare et al., 2016; Ostrovski et al., 2017).

For RANDOM, we implement the two networks with 64-64 MLP. For CURIOSITY and RIDE, we use 64-64 MLP for state embedding model, forward dynamics model and inverse dynamics model, respectively. However, with extensive hyperparameters search, we are not able to reproduce the RIDE results in MiniGrid environments even with the exactly same neural architecture and hyperparameters as suggested in (Raileanu & Rocktäschel, 2020). Thus, we use the authors' implementation[2] (Raileanu & Rocktäschel, 2020), which is well-tuned on MiniGrid tasks. Our reported result

---

[1] https://github.com/openai/baselines
[2] https://github.com/facebookresearch/impact-driven-exploration

is on par with that reported in (Raileanu & Rocktäschel, 2020). For SIL and AMIGO, we use the implementations from the authors[34] with the recommended hyperparameters.

For the methods based on intrinsic rewards, i.e., RIDE,, AMIGO CURIOSITY, RANDOM, and COUNT, we search intrinsic reward coefficient from $\{0.1, 0.05, 0.01, 0.005, 0.001, 0.0005, 0.0001\}$. For RIDE, the intrinsic reward coefficient is $0.1$ for MiniGrid-KeyCorridorS2R3-v0, MiniGrid-KeyCorridorS3R3-v0, MiniGrid-KeyCorridorS4R3-v0, MiniGrid-MultiRoom-N10S4-v0, MiniGrid-MultiRoom-N7S4-v0 and MiniWorld-Maze-S5, and $0.5$ for MiniGrid-MultiRoom-N7S8-v0, MiniGrid-MultiRoom-N10S10-v0 and MiniGrid-MultiRoom-N12S10-v0. For AMIGO and CURIOSITY, the intrinsic reward coefficient is $0.1$ for all the environments. For COUNT, the intrinsic reward coefficient is $0.005$ for all the environments. For RANDOM, the intrinsic reward coefficient is $0.001$ for all the environemnts. We also tune the entropy coefficient of RIDE from $\{0.001, 0.0005\}$, as suggested by the original paper. The entropy coefficient is $0.0005$ for MiniGrid-KeyCorridorS2R3-v0, MiniGrid-KeyCorridorS3R3-v0, MiniGrid-KeyCorridorS4R3-v0, MiniGrid-MultiRoom-N10S4-v0, MiniGrid-MultiRoom-N7S4-v0 and MiniWorld-Maze-S5, and $0.001$ for MiniGrid-MultiRoom-N7S8-v0, MiniGrid-MultiRoom-N10S10-v0 and MiniGrid-MultiRoom-N12S10-v0.

The $w_0$, $w_1$, and $w_2$ are selected as follows. We first run RAPID on MiniGrid-MultiRoom-N7S8-v0. We choose this environment because it is not too hard nor too easy, so we expect that it is representative among all the considered MiniGrid environments. We fix $w_0$ to be $1$, and select $w_1$ and $w_2$ from $\{10^{-4}, 10^{-3}, 10^{-2}, 10^{-1}, 10^0, 10^1, 10^2, 10^3, 10^4\}$, that is, there are $9 \times 9 = 81$ combinations. Note that we can fix $w_0$ because only the relative ranking matters in the ranking buffer. The selected $w_1$ and $w_2$ are then directly used in other environments without tuning. We further conduct a sensitivity analysis on MiniGrid-MultiRoom-N7S8-v0 in Figure 8. We observe that RAPID has good performance in a very wide range of hyperparameter choices.

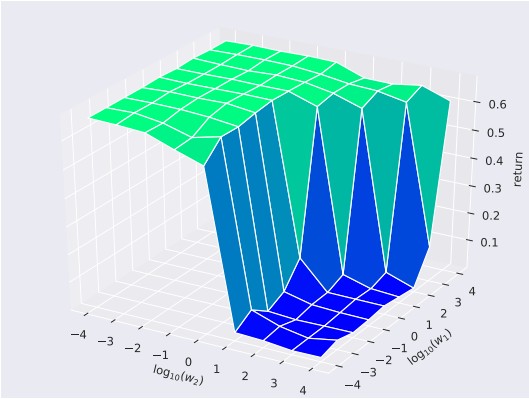

Figure 8: Sensitivity analysis of $w_1$ and $w_2$ ($w_0$ is fixed to 1) in MiniGrid-MultiRoom-N7S8-v0. All the experiments are run $3 \times 10^6$ timesteps. Note that $3 \times 10^6$ is more than enough for RAPID to converge in this environment (see Figure 4). The average results over 5 independent runs are plotted.

# B    ENVIRONMENTS

## B.1    MINIGRID ENVIRONMENTS

MiniGrid[5] is a suite of light-weighted and fast gridworld environments with OpenAI gym interfaces. The environments are partially observable with a grid size of $N \times N$. Each tile in the gird can have at most one object, that is, 0 or 1 object. The possible objects are wall, floor, lava, door, key, ball, box and goal. Each object will have an associated color. The agent can pick up at most one object (ball or key). To open a locked door, the agent must use a key. Most of the environments in

---

[3]https://github.com/junhyukoh/self-imitation-learning
[4]https://github.com/facebookresearch/adversarially-motivated-intrinsic-goals
[5]https://github.com/maximecb/gym-minigrid

MiniGrid are procedurally-generated, i.e. a different grid will be sampled in each episode. Figure 9 shows the grids in four different episodes. The procedurally-generated nature makes training RL challenging because the RL agents need to learn the skills that can generalize to different grids. The environments can be easily modified so that we can create different levels of the environments. For example, we can modify the number of rooms and the size of each room to create hard-exploration or easy-exploration environments. The flexibility of MiniGrid environments enables us to conduct systematic comparison of algorithms under different difficulties.

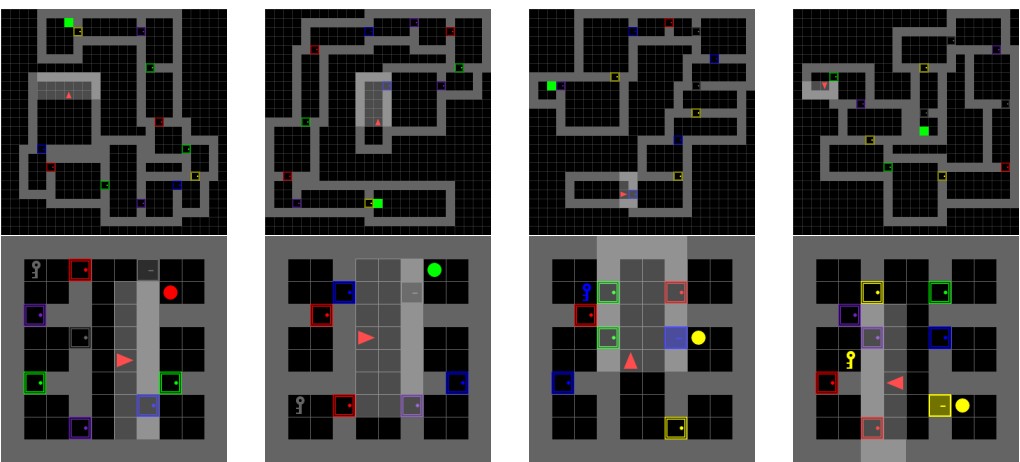

Figure 9: Rendering of MultiRoom-N12-S10 (top row) and KeyCorridor-S4-R3(bottom row) in 4 different episodes. The environments are procedually-generated, i.e., a different room is generated in a new episode.

The original observations of MiniGrid are dictionaries, which consist of a *image* field providing a partially observable view of the environment, and a *mission* field describing the goal with texts. In our experiments, we use `ImgObsWrapper` which only keeps the *image* field. The environment uses a compact encoding with 3 input values per visible grid cell. Note that the cells behind the wall or unopened doors are invisible.

The possible actions in MiniGrid are (1) turn left, (2) turn right, (3) move forward, (4) pick up an object, (5) drop the carried object, (6) open doors/interact with objects, and (7) done. The agent will remain in the same state if the action is not legal. For example, if there is no object in front of the agent, the agent will remain in the same state when performing action (4) pick up an object.

We focus on MultiRoom-NX-SY and KeyCorridor-SX-RY, where X and Y are hyperparameters specifying the number of rooms or the room sizes in the environments. With larger X and Y, the environments will become more difficult to solve due to the sparse rewards. Figure 10 shows the environments (with different difficulties) in this work. In MultiRoom-NX-SY, the agent needs to navigate the green tile in the last room. If the agent successfully navigates the goal, the agent will receive a reward of 1 subtracting some penalties of the timesteps spent. In KeyCorridor-SX-RY, the agent needs to pick up the key, use the key to open the locked door, and pick up the ball. Similarly, the agent will receive a reward of 1 subtracting some penalties of the timesteps spent if it picks up the ball. Both environments are extremely difficult to solve using RL alone due to the sparse rewards.

## B.2 MINIWORLD MAZE ENVIRONMENT

MiniWorld[6] is a minimalistic 3D interior environment simulator. Similar to Minigrid, we can easily create environments with different levels. We focus on MiniWorld-Maze, where the agent is asked to navigate the goal through a procedurally-generated maze. The agent has a first-person partially observable view of the environment. This environment is extremely difficult to solve due to sparse reward, long-horizon, and the procedurally-generated nature of the environments.

---

[6] https://github.com/maximecb/gym-miniworld

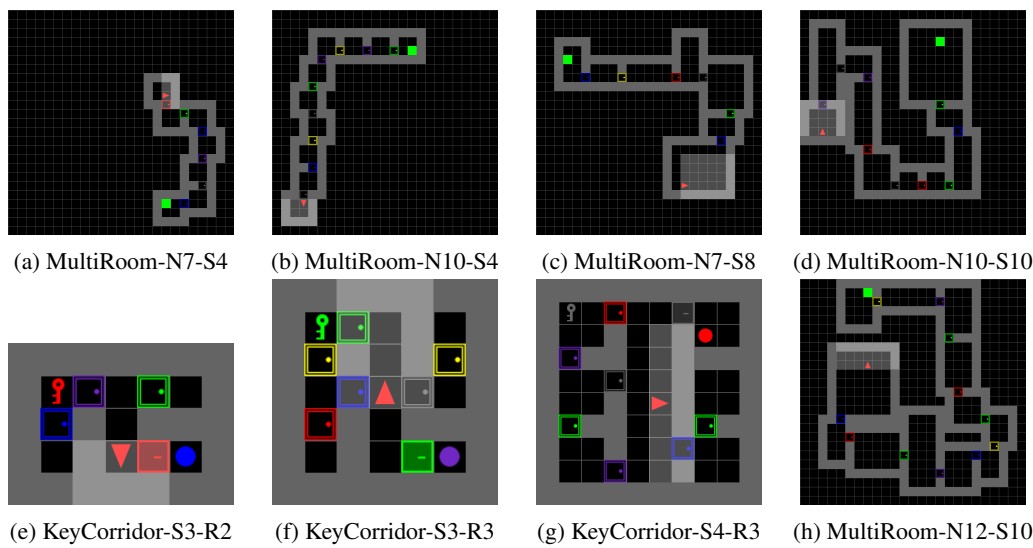

| (a) MultiRoom-N7-S4 | (b) MultiRoom-N10-S4 | (c) MultiRoom-N7-S8 | (d) MultiRoom-N10-S10 |
|---|---|---|---|
| (e) KeyCorridor-S3-R2 | (f) KeyCorridor-S3-R3 | (g) KeyCorridor-S4-R3 | (h) MultiRoom-N12-S10 |

Figure 10: Rendering of Minigird environments used in the work.

In this work, we focus on MiniWorld-Maze-S5, a variant of the MiniWorld Maze environment with $5 \times 5$ tiles. Figure 11 shows the top view of the mazes in four different episodes. In each episode, the agent and the goal are initialized in the top-left and bottom-right corners of the map, respectively. In this way, we ensure that the agent and the goal are far away enough so that the agent can not easily see the goal without exploration. A random maze will be generated in each episode. There are three possible actions: (1) move forward, (2) turn left, and (3) turn right. The agent will move $0.4 \times tile\_length$ if moving forward, where $tile\_length$ is the length of one side of a tile. The agent will rotate 22.5 degrees if turning right/left. The time budget of each episode is 600. The agent will not receive any positive reward if it can not navigate the goal under the time budget.



Figure 11: Example generated mazes in 4 diffident episodes.

### B.3    SPARSE MUJOCO ENVIRONMENTS

Mujoco is a physical engine for continuous control (Todorov et al., 2012). Figure 12 shows the rendering of the four MuJoCo tasks used in our work. The original MuJoCo environments use dense rewards, i.e., a well-defined reward is given in each timestep. However, in many scenarios, well-defined dense rewards may be not available. We consider a variant of MuJoCo task using only episodic reward. Specifically, the original rewards are accumulated and only an episodic reward is given in the final timestep. The agent will receive a 0 reward in all the other timesteps. This sparse setting makes the tasks much more challenging to solve. Contemporary RL algorithms will usually suffer from the sparse rewards and deliver unsatisfactory performance. We aim to use these sparse variants to study whether our RAPID can effectively capture the sparse rewards.

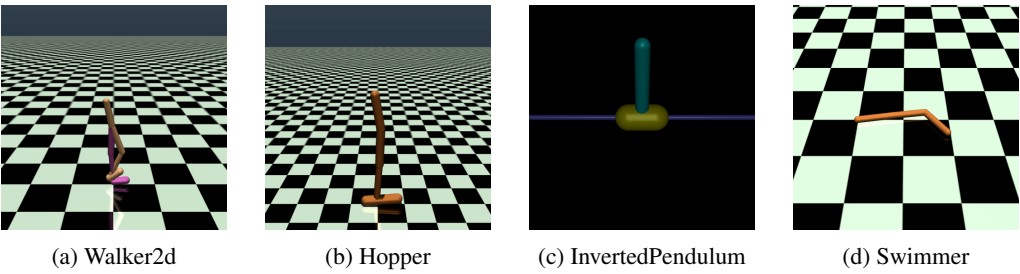

(a) Walker2d    (b) Hopper    (c) InvertedPendulum    (d) Swimmer

Figure 12: Rendering of Mujoco environments.

## C    ABLATIONS

Figure 13 shows the learning curves of RAPID against different ablations. Without local scores, we observe significant performance drop on challenging environments in terms of sample efficiency and final performance, i.e., on MultiRoom-N7-S8, MultiRoom-N10-S10, MultiRoom-N12-S10, KeyCorridor-S3-R3, and KeyCorridor-S4-R3. This suggests that the proposed local score plays an important role in encouraging exploration. We also observe that without buffer, the agent fails to learn a good policy. Naively using the proposed scores as intrinsic reward will harm the performance. Using the extrinsic reward and the global score to rank the episodes also contribute in the challenging environments, such as KeyCorridor-S4-R3. Therefore, the three proposed scoring methods may be complementary and play different roles in different environments.

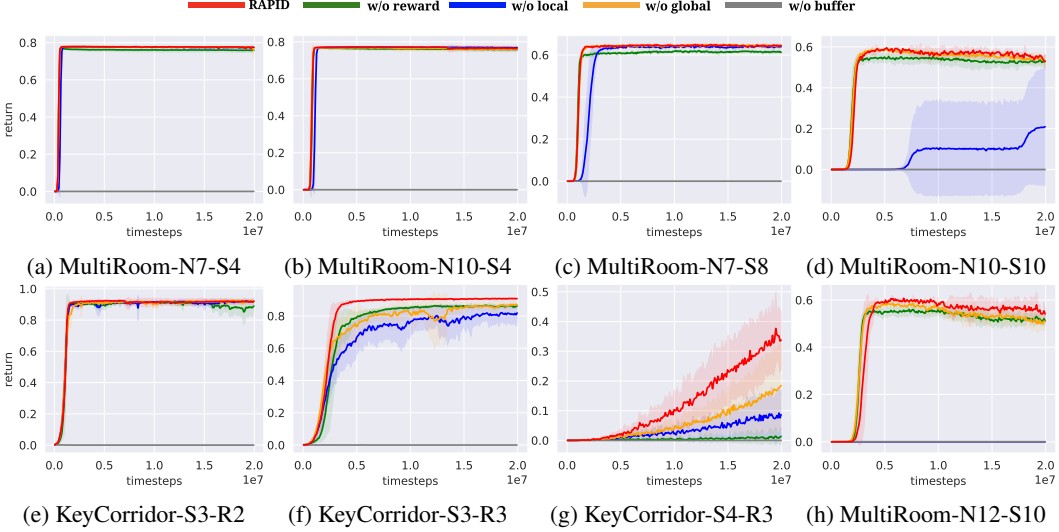

(a) MultiRoom-N7-S4  (b) MultiRoom-N10-S4  (c) MultiRoom-N7-S8  (d) MultiRoom-N10-S10

(e) KeyCorridor-S3-R2  (f) KeyCorridor-S3-R3  (g) KeyCorridor-S4-R3  (h) MultiRoom-N12-S10

Figure 13: Learning curves of RAPID and the ablations

## D    FULL ANALYSIS RESULTS

### D.1    IMPACT OF THE NUMBER OF UPDATE STEPS

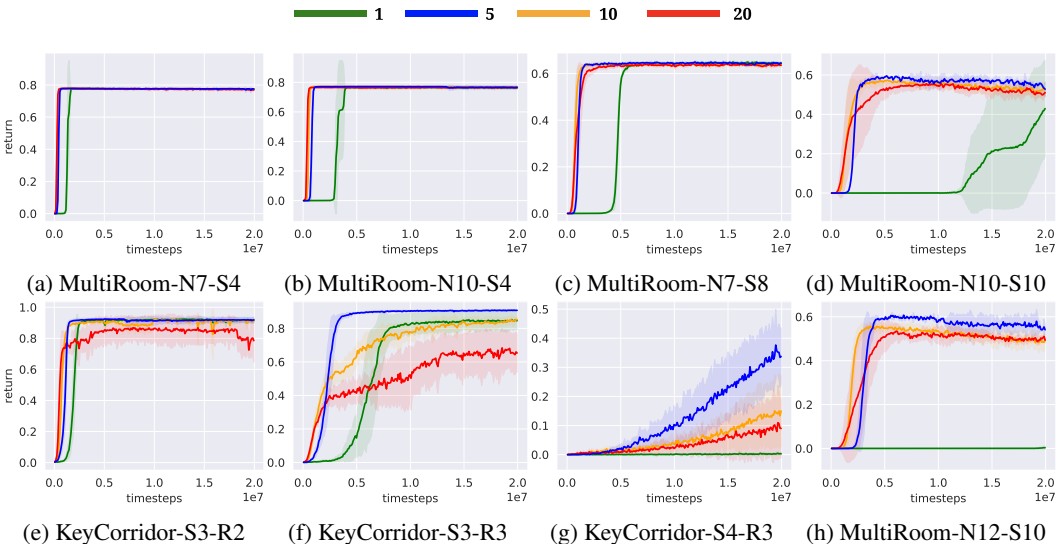

Figure 14: Impact of training steps $S$ on MiniGrid environments.

## D.2 IMPACT OF BUFFER SIZE

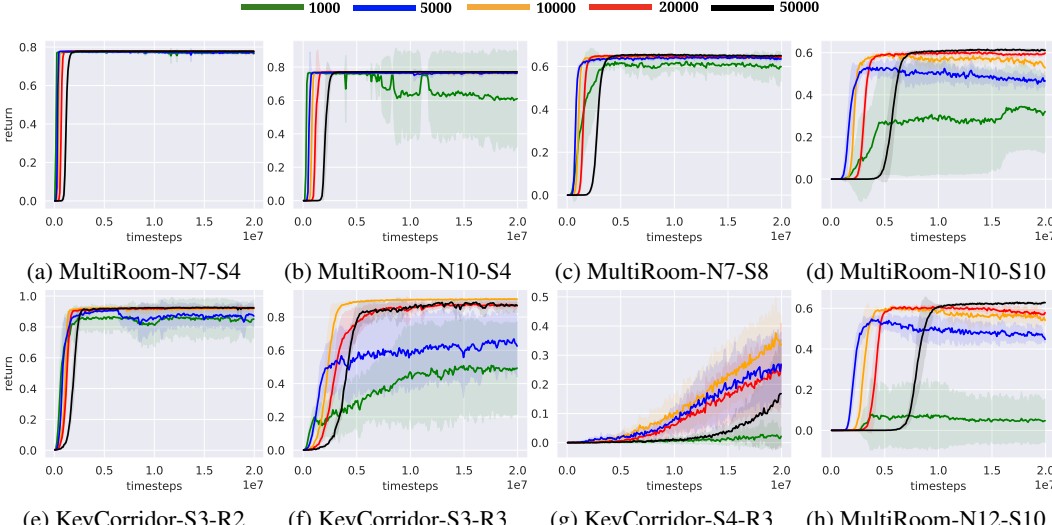

Figure 15: Impact of buffer size $D$ on MiniGrid environments.

### D.3 PURE EXPLORATION ON MINIGRID

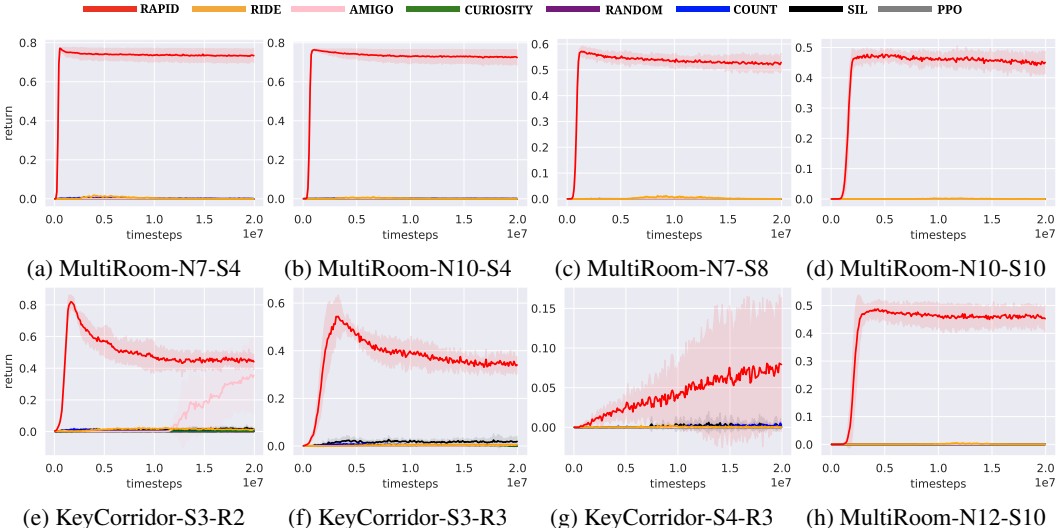

Figure 16: Extrinsic rewards achieved by RAPID and baselines with pure exploration

### D.4 LOCAL EXPLORATION SCORE OF PURE EXPLORATION ON MINIGRID

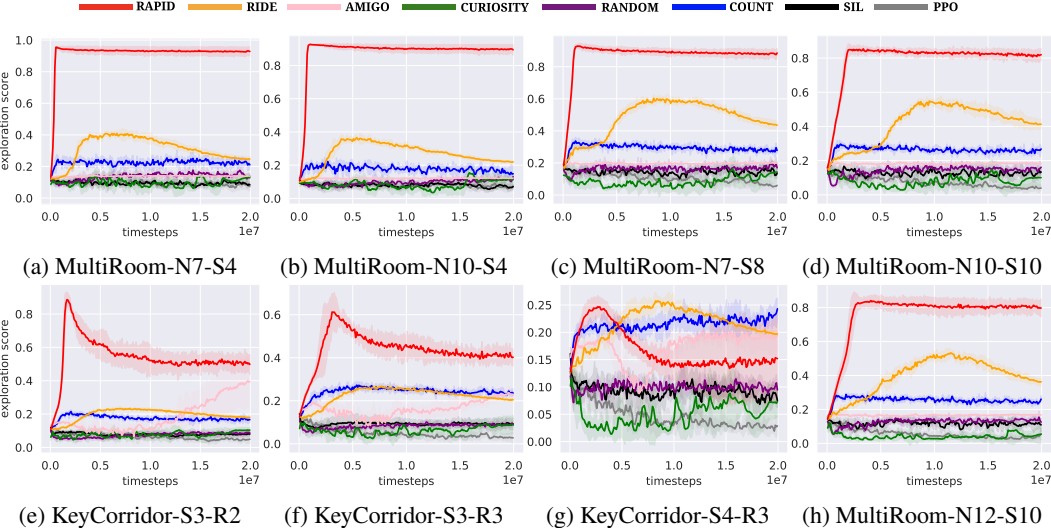

Figure 17: Local exploration scores achieved by RAPID and baselines with pure exploration

# E    LEARNING CURVES WITH MORE ROOMS

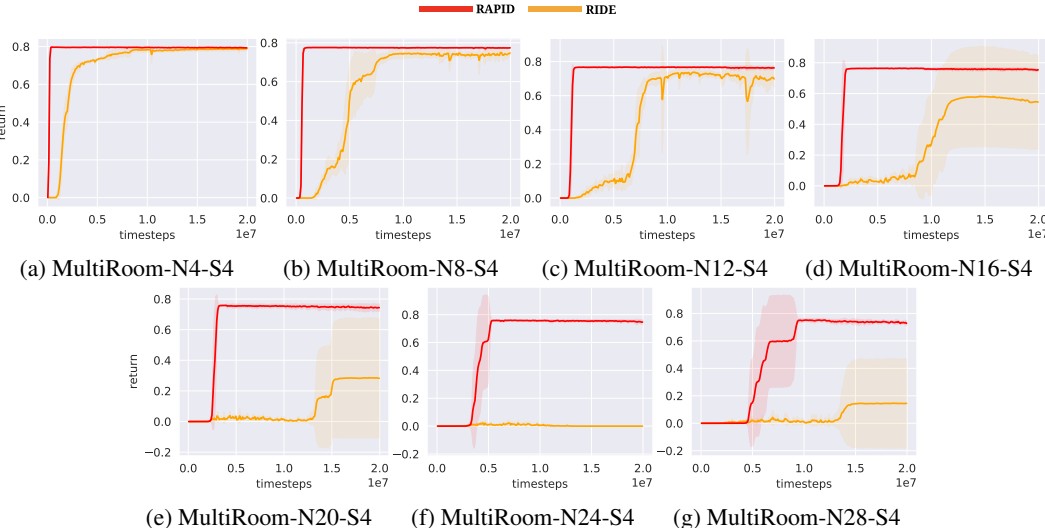

(a) MultiRoom-N4-S4    (b) MultiRoom-N8-S4    (c) MultiRoom-N12-S4    (d) MultiRoom-N16-S4

(e) MultiRoom-N20-S4    (f) MultiRoom-N24-S4    (g) MultiRoom-N28-S4

Figure 18: The learning curves with more rooms

# F    LEARNING CURVES WITH LARGE ROOMS

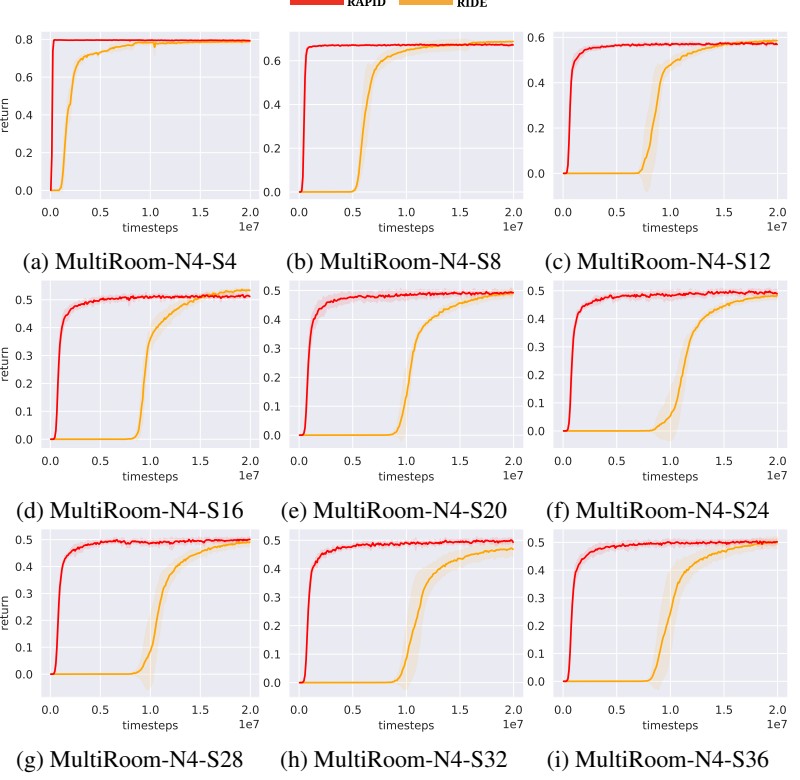

(a) MultiRoom-N4-S4    (b) MultiRoom-N4-S8    (c) MultiRoom-N4-S12

(d) MultiRoom-N4-S16    (e) MultiRoom-N4-S20    (f) MultiRoom-N4-S24

(g) MultiRoom-N4-S28    (h) MultiRoom-N4-S32    (i) MultiRoom-N4-S36

Figure 19: The learning curves with larger rooms sizes.

# G    ABLATION AND ANALYSIS OF MINIWORLD-MAZE

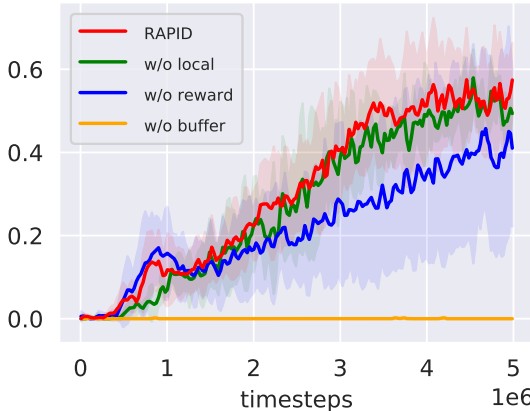

Figure 20: Learning curves of RAPID and the ablations on MiniWorld Maze. We observe minor performance drop when removing the local score, substantial performance drop when removing extrinsic rewards or the buffer.

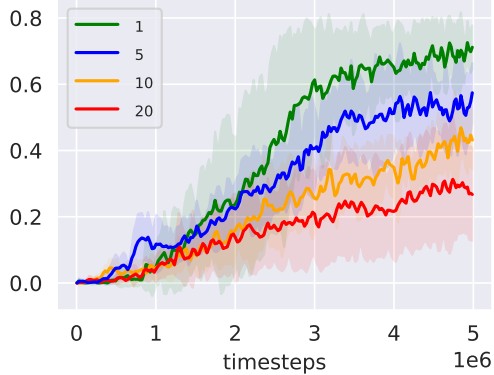

Figure 21: Impact of training steps on MiniWorld Maze.

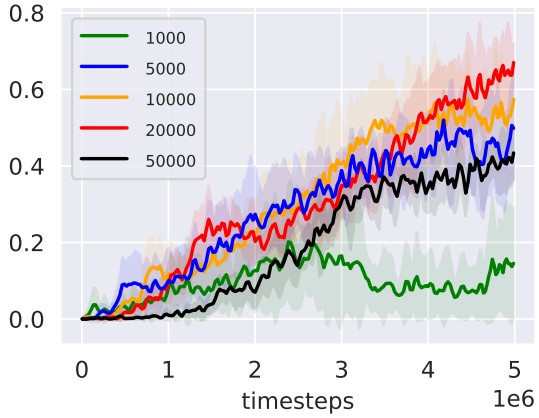

Figure 22: Impact of buffer size on MiniWorld Maze.

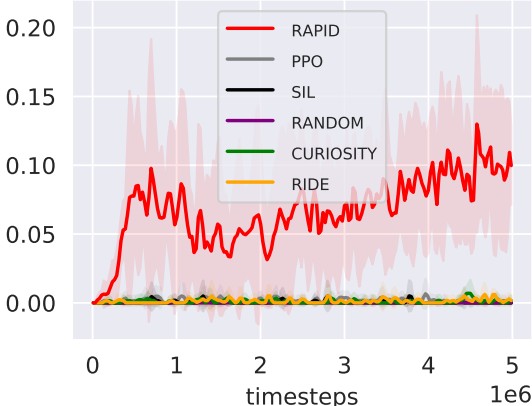

Figure 23: Extrinsic rewards achieved by RAPID and baselines on MiniWorld Maze with pure exploration.

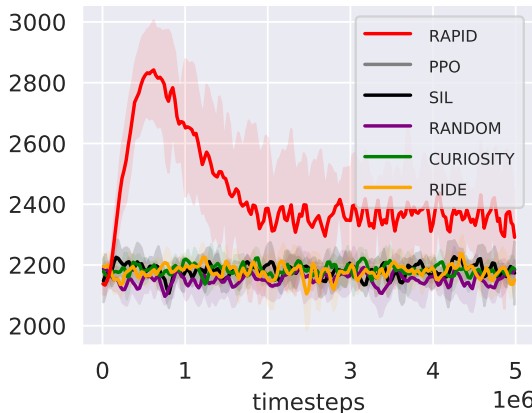

Figure 24: Local exploration scores achieved by RAPID and baselines on MiniWorld Maze with pure exploration.

## H  COMPARISON WITH STORING ENTIRE EPISODES IN THE BUFFER

A variant of RAPID is to keep the entire episodes in the buffer instead of state-action pairs. The intuition of keeping the whole episode is that it is possible that a particular state-action pair may only be good in terms of exploration, in the context of the rest of the agent's trajectory in that episode. To test this variant, we force the buffer to keep the entire episode by allowing the episode at the end of the buffer to exceed the buffer size. For example, given that the buffer size is 5000, if the length of the end episode is 160 and the number of state-action pairs in the buffer is 5120, we do not cut off the end episode and exceptionally allow the buffer to store 5120 state-action pairs. In this way, we ensure that the entire episodes are kept in the buffer. We run this experiment on MiniGrid-MultiRoom-N7-S8-v0 (Figure 25). We do not observe a clear difference in the learning curves.

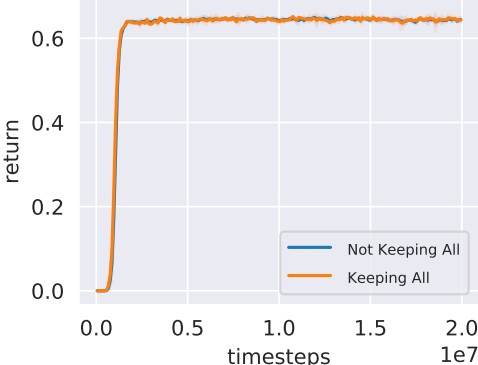

Figure 25: Comparison of keeping all the state-action pairs of an episode and not keeping all the state-action pairs (i.e., a fixed buffer size) on MultiRoom-N7-S8. The experiments are run 5 times with different random seeds. We observe no clear difference in the learning curves of these two implementations.

## I  RESULTS ON PROCEDURALLY-GENERATED SWIMMER

The standard MuJoCo environments are singleton. Here, we modify the Swimmer-v2 environment to make it procedurally-generated. Specifically, we make the environment in each episode different by modifying the XML configuration file of MuJoCo engine in every new episode. We consider two environmental properties, i.e., density and viscosity. Density is used to simulate lift and drag forces, which scale quadratically with velocity. Viscosity is used to simulate viscous forces, which scale linearly with velocity. In the standard Swimmer-v2 environment, the density and the viscosity are fixed to 4000 and 0.1, respectively. In the first experiment, we uniformly sample the density in $[2000, 4000]$ in each new episode to make it procedurally-generated, denoted as Density-Swimmer. Similarly, in the second environment, we uniformly sample the velocity in $[0.1, 0.5]$, denoted as Velocity-Swimmer. These two variants make RL training more challenging since the agent needs to learn a policy that can generalize to different densities and velocities. Similarly, we accumulate the rewards to the final timestep of an episode to make the environments sparse. Both environments are provided in our code for reproducibility. The results are reported in Figure 26. We observe that all methods learn slower on these two variants. For example, RAPID only achieves around 190 return in Density-Swimmer, while RAPID achieves more than 200 return in Swimmer. Similarly, SIL reaches around 100 return after around $5 \times 10^6$ timesteps in Density-Swimmer, while it achieves around 100 return after around $3 \times 10^6$ timesteps in Swimmer. Nevertheless, RAPID can discover very good policies even in these challenging procedurally-generated settings.

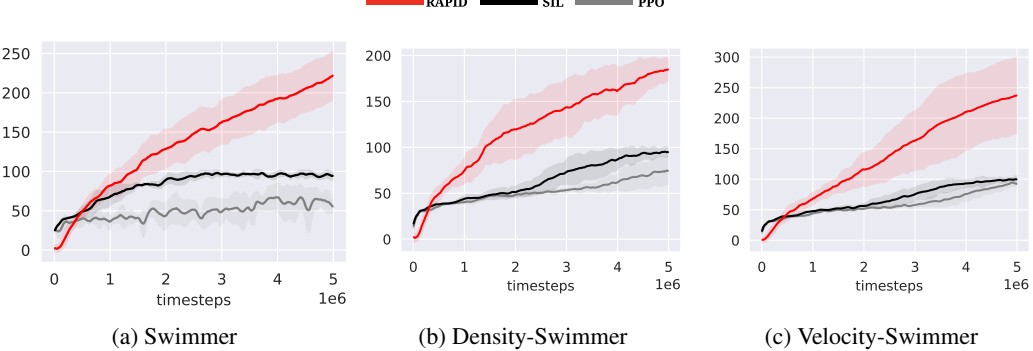

(a) Swimmer      (b) Density-Swimmer      (c) Velocity-Swimmer

Figure 26: (a) is the singleton swimmer environment. In (b) the density is procedurally-generated. In (c) the velocity is procedurally-generated. All the methods tend to learn slower in the procedurally-generated settings. Nevertheless, RAPID is still able to discover good policies in these challenging variants.

## J  DISCUSSIONS OF ANNEALING

In our experiments, we find that annealing the updating step to $0$ sometimes helps improve the sample efficiency. However, in some environments, annealing will have a negative effect on the final performance. To show the effect of annealing, we plot learning curves with or without annealing on MiniGrid-KeyCorridorS3R2-v0 and MiniGrid-MultiRoom-N12-S10-v0 in Figure 27. We observe that annealing can help in KeyCorridor-S3-R2. The possible explanation is that, after the agent navigates the goal, it has access to the reward signal. the PPO objective is sufficient to train a good policy with the extrinsic rewards. The imitation loss may harm the performance since it could be better to perform exploitation rather than exploration at this stage. However, in MultiRoom-N12-S10, annealing will harm the performance. In particular, the performance drops significantly in the final stage when the updating frequency of imitation loss approaches zero. The possible reason is that MultiRoom-N12-S10 needs very deep exploration to find the goal. Even though the PPO agent can navigate the goal, it may easily suffer from insufficient exploration when it encounters a new environment. As a result, the agent needs to keep exploring the environment in order to generalize. Introducing imitation loss will ensure that the agent keeps exploring the environments.

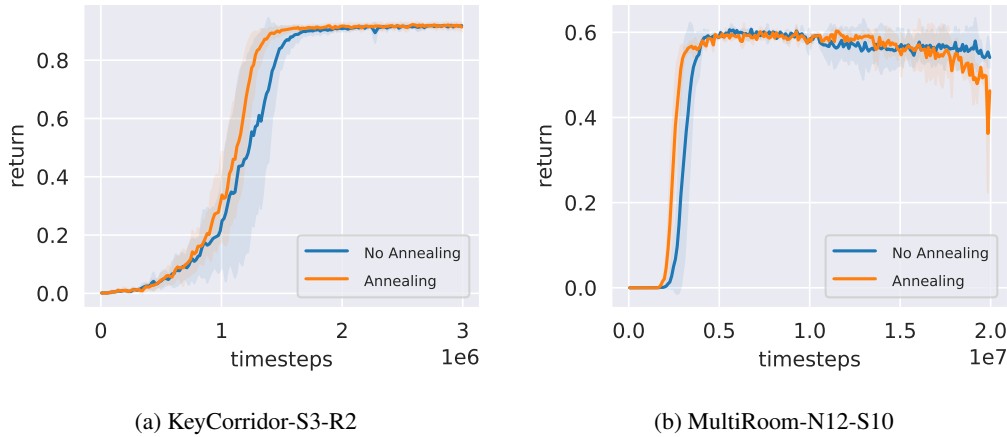

(a) KeyCorridor-S3-R2      (b) MultiRoom-N12-S10

Figure 27: Annealing versus not annealing. Annealing is helpful in KeyCorridor-S3-R2. However, the policy fails to converge with annealing in MultiRoom-N12-S10.

