# OpenReview forum: "Rank the Episodes: A Simple Approach for Exploration in Procedurally-Generated Environments"
_ICLR.cc/2021/Conference — ICLR 2021 Poster_

### Official Review · AnonReviewer3 · 2020-10-28
**Official Blind Review #3**

**Rating:** 6
**Confidence:** 3

**Review:**

This paper presents an exploration method for procedurally-generated environments, RAPID, which imitates the past episodes that have a good exploration behavior. First, authors introduce exploration scores, local score for per-episode view of the exploration behavior, and global score for long-term and historical view of exploration. The authors use the weighted sum of these two exploration scores and extrinsic reward as a final episodic exploration score. They rank the state-action pairs based on episodic exploration score and train the agent to imitate behaviors with high score. In experiments, they show the results by comparing state-of-the-art algorithms in several procedurally-generated environments.

This paper is well written, but I have following concerns and questions about the paper.
- MiniGrid seems to be a partially observable problem in which only 7x7 observation is given to the agent. What does the state used when calculating the exploration score in the paper mean?
- Moreover, the paper also includes experiments on problems with continuous state space. In that case, I wonder the details of how the number of states and distinct states are handled.
- When calculating the episodic exploration score (Equation (3)), it is thought that the results vary greatly depending on the weight $w_0$ - $w_2$ values. In the paper, it seems necessary to show the criteria for determining the weights and the difference in results according to the weights.
- In order to show the effectiveness of the ranking buffer, in addition to show with/without results as shown in Table 1, the results when using the replay buffer without considering the ranking should be compared together.
- It would be better if the MuJoCo tasks in Figure 7 could be explained more clearly. (ex. The existing MuJoCo would be a singleton, but what was changed to make it a procedurally-generated environment?)
- In the experiment, the explanation is mainly based on KeyCorridor, but there is no explanation for what task it is. It would be better if a brief explanation of the KeyCorridor task is included in the paper (with the reason why this task is hard).
- The author claims that the proposed method is effective in very sparse environments, but I am not sure exactly what makes it effective in very sparse environments. The justification for this claim needs to be explained more intuitively.

This paper contains experimental results for various tasks, but It is unclear the contribution of this paper and the reason why the proposed method is effective in very sparse and procedurally-generated environments. It seems necessary to explain intuitive motivation more clearly in the paper. Moreover, for the ablation study, It also needs precise verification that can show the effect of each component more intuitively, not ablation that simply separates and compares each component.

---

> ### Author Response · Authors · 2020-11-17
> **Thank you for the criticisms and the time!**
>
> We thank the reviewer for the criticisms and the time. Please find our response and some clarification as follows.
>
> **MiniGrid seems to be a partially observable problem in which only 7x7 observation is given to the agent. What does the state used when calculating the exploration score in the paper mean?**
>
> We flatten 7x7 observation as the state vector and obtaining the scores by counting the state. In MiniGrid, the state is discrete, so counting is meaningful.
>
> **Moreover, the paper also includes experiments on problems with continuous state space. In that case, I wonder the details of how the number of states and distinct states are handled.**
>
> For continuous state space, we only use the local score. The local score is obtained by the standard deviation. In the updated version, we have formally defined the local score in continuous state space in Eq. 2. We have also provided details of how to handle the state for all the environments in Table 3 of Appendix A.
>
> **When calculating the episodic exploration score (Equation (3)), it is thought that the results vary greatly depending on the weight values. In the paper, it seems necessary to show the criteria for determining the weights and the difference in results according to the weights.**
>
> Thank you for pointing this out. We grid-searched the weights on Multi-Room-N7-S8 and adopted the selected weights in other MiniGrid environments without further tuning. In the updated version, the details are provided in Appendix A. We also provide a sensitivity analysis of the weights in Figure 8.
>
> **In order to show the effectiveness of the ranking buffer, in addition to show with/without results as shown in Table 1, the results when using the replay buffer without considering the ranking should be compared together.**
>
> Without the ranking, the buffer will reduce to ring buffer. The ring buffer will not provide any selection of good exploration behaviors. Thus, conducting imitation learning is simply reproducing all past experiences. Since there is no selection if using a ring buffer, we expect that the ring buffer will not help exploration. To address your concern, we have run the variant without ranking and added the result to Table 1. As expected, the agent does not perform well without ranking.
>
> **It would be better if the MuJoCo tasks in Figure 7 could be explained more clearly. (ex. The existing MuJoCo would be a singleton, but what was changed to make it a procedurally-generated environment?)**
>
> All the environments in Figure 7 are singleton. We designed these experiments solely to test whether RAPID can handle continuous state/action spaces. To address your concern, in the updated version, we make the Swimmer procedurally-generated. Specifically, we sample a different value of the physical property in each new episode. In this setting, the agent needs to learn a policy that can generalize to different parameterized physical worlds. The results are presented in Appendix I. We observe that RAPID can still discover very good policies in the procedurally-generated setting. We have explicitly clarified that these environments are singleton in Section 3.4 of the updated version. The new environments have been merged in our codebase and will be open-sourced for reproducibility.
>
> **In the experiment, the explanation is mainly based on KeyCorridor, but there is no explanation for what task it is. It would be better if a brief explanation of the KeyCorridor task is included in the paper (with the reason why this task is hard).**
>
> Thanks for the feedback! We have added some descriptions in Section 3.1 to discuss why these environments are hard. Due to the space limit, we put more detailed descriptions in Appendix B.1.
>
> **The author claims that the proposed method is effective in very sparse environments, but I am not sure exactly what makes it effective in very sparse environments. The justification for this claim needs to be explained more intuitively.**
>
> The key idea is in the episodic scores. For example, the local score quantifies how many distinct states there are in one episode. Even though there is no extrinsic reward, the agent learns to visit as many distinct states as possible, which encourages exploration since the agent is asked not to visit the same state. RAPID decouples exploration into local and global scores. The local score is a per-episode view and is generalization-aware. The global score provides a long-term view. To make use of the highly-scored data, we design a ranking buffer to maximize the scores by imitating the good exploration experiences. Please let us know if you have any specific questions about the intuitions of any parts of our algorithm.

---

> > ### Comment · AnonReviewer3 · 2020-11-24
> > **Response to rebuttal**
> >
> > Thank you for taking the time to clarifications and considering my comments. All of my questions and concerns were resolved by rebuttal and revision. This seems well presented with good results. I will raise my score.

---

### Official Review · AnonReviewer4 · 2020-10-29
**Good approach to exploration in procedurally generated environments, may not extend pas tabular environments.**

**Rating:** 7
**Confidence:** 3

**Review:**

This paper presents RAPID, an exploration algorithm for procedurally generated environments. The paper introduces an exploration scores composed of a local and global score. The local score is computed per-episode, it is the fraction of distinct states visited during an episode, the global score keeps track of the exploratory effort of the agent over the whole training procedure.
These two scores are then added to the environment reward and used to improve the agent's policy with imitation learning. RAPID is shown have a faster convergence and lead to a higher performance than previous exploration algorithms for procedurally generated environments.

I found the paper well written and easy to follow. The authors provided many explanations and figures to illustrate the behaviour of their algorithm.
The experiment section was great, it included many experiments on different environments as well as an extensive ablation study to showcase the benefits of RAPID. I think the key insight of this paper is using the local score as a reward for exploration and using it with imitation learning. It seems to be a great proxy for counts in the setting of procedurally generated games in the tabular case. I was actually surprised by the performance of the local score, as it is a per-episode metric it seemed to be a weak signal and that would be hard to learn from it. I'm guessing that the ranking of episodes is key here to make the algorithm work.

On the other hand I was not convinced by the utility of the global score. First of all it is not clear if the global score is state dependent or not, I assume it is not. The authors say: "the basic building blocks (e.g., walls, doors, etc.), the number of rooms, and the size of each room will be similar. From a global view, the agent should explore the regions that are not well explored in the past", unfortunately while similar states may be encountered they will likely not be equal, a metric defined using raw counts will often not be useful.
"From a global view, the agent should explore the regions that are not well explored in the past" This sentence hints at the notion that should be more clearly discussed: generalization. While the local score only quantifies exploration within a single episode, the global score tries to leverage past experience to improve exploration, however to do so as the environment changes constantly some kind of generalization is necessary. I am not sure how a metric that is not aware of the generalization can be helpful. This is also shown in the experiment section, the global score that does not appear to significantly contribute to performance except on KeyCorridor-S4-R3 (any reason why?) and it is even disabled for Mujoco experiments.
Another issue is that contrary to some other algorithm like RIDE, RAPID does not easily extend to environments with large or continuous state action spaces because obtaining meaningful counts in these environments is difficult, to be fair this an issue still faced in non procedurally generated environments. It would also be nice to know how RAPID was applied to Mujoco experiments, was the observation space discretized?

Overall I lean towards acceptance as I think that the paper presented a good idea with some solid experiments even though I have doubts regarding how applicable it can be to more complex environments.

Few typos:
the methods based on intrinsic rewards -> methods based on intrinsic rewards
may take extremely long time -> an extremely long time
exploration in sparse environments. -> environments with sparse rewards
could vary in diffident environments -> different environments

---

> ### Author Response · Authors · 2020-11-17
> **Thank you for the insightful feedback!**
>
> We thank the reviewer for the insightful feedback and the time. Please find our response as follows.
>
> **I am not sure how a metric that is not aware of the generalization can be helpful.**
>
> Yes, the global score is not aware of the generalization. The generalization part is in the local score since it is computed in a per-episode view. The global score is designed to model the information that has not been changed in generalization. We believe whether there exists such unchanged information depends on what environment is used. While the global score seems to be not very effective in most MiniGrid environments, in certain cases (KeyCorridor-S4-R3), it helps a lot. A possible reason is that the local score is a too weak signal in KeyCorridor-S4-R3. The global score may help alleviate this issue since it provides some historical information. Besides, using the global score at least does not harm the performance according to our results. We believe whether and under what circumstances the global score is helpful depends on the environment. It is possible that some environments do not need a global score at all. In this case, we can simply set $w_2 = 0$ to remove the global score. We have added the above discussions to the paragraph of RQ2 in the updated version.
>
> **it is even disabled for Mujoco experiments**
>
> In our experiments, we disabled all the global scores in continuous state space because counting continuous state is meaningless. Pseudo-counts could be used to address this issue, which we will try in our future work.
>
> **Another issue is that contrary to some other algorithm like RIDE, RAPID does not easily extend to environments with large or continuous state action spaces because obtaining meaningful counts in these environments is difficult, to be fair this an issue still faced in non procedurally generated environments.**
>
> For the local score, we calculate the standard deviations of states, as defined in Eq. 2 in the updated version. For the global score, it is possible to use pseudo-count, which we will try in our future work. While MiniGrid environments are discrete, the observation of MiniWorld Maze is 60*80 images. In MuJoCo, both the state and action spaces are continuous. For larger state space, one possible extension of RAPID could be learning the state embedding and calculating the scores on the embeddings, which we will try in the future on more complicated environments.
>
> **It would also be nice to know how RAPID was applied to Mujoco experiments, was the observation space discretized?**
>
> We use the standard deviation to quantify the local score. In the updated version, this score is formally defined in Eq. 2. We have also summarized the calculation of the scores in Table 3, Appendix A.
>
> **Few typos**
>
> Thank you! We have fixed all of them.

---

> > ### Comment · AnonReviewer4 · 2020-11-25
> > **Response to rebuttal**
> >
> > Thank you for your response and the updated paper. I particularly appreciate the additional experiments and the discussion in section 5. The additional information provided on Mujoco experiments is also helpful to understand how RAPID can be applied to continuous state action spaces.
> > I'm raising my score to accept.

---

### Official Review · AnonReviewer2 · 2020-10-29
**Review 1**

**Rating:** 7
**Confidence:** 4

**Review:**

Summary: This paper focuses on building reinforcement learning agents for procedurally generated environments. In particular it presents a method RAPID to intrinsically reward previously seen *good* exploration behaviors by attempting to imitate them in the future. Strong results are presented across multiple procedurally generated environments.

Pros:
1. Procedurally generated environments provide a rigorous test of an agent's ability to generalize as exploration strategies learned cannot be reused as a whole. To counter this, they operate on the hypothesis that high rewarding trajectories are more likely to contain more generalizable exploration behaviors. The break down of this reward into local (per episode) and global extrinsic rewards gives the agent some idea of areas of the game it historically performs poorly on.
2. Given that the idea itself is simple (yet effective!), I appreciate that most of the main paper is dedicated to experiments - with each set of experiments designed to answer a particular research question. This make it easily accessible and digestible - answering many of the doubts I had when reading through the intro and methodology sections.

Cons: I have a couple of primary concerns mostly dealing with the underlying assumptions made.
1. This method rewards good exploration behavior historically and as such depends on the ability of the agent to discover such trajectories in the first place.
2. High extrinsic reward implies good exploration behavior.
Both of these assumptions likely do not hold in environments with sparse/ill-placed/deceptive rewards, challenges that environments such as Montezuma's Revenge/Net Hack/Interactive Fiction games are notorious for. In such environments, it is not only difficult to discover good exploration behaviors but also the overall extrinsic rewards themselves often lead to dead ends and globally suboptimal trajectories. The global score, being effectively count-based, alleviates some of this, but it is likely that this form of intrinsic that is based on extrinsic reward will likely fail in such scenarios. It would strengthen the paper to see positive results in such an environment.

Minor: A few of citations to the ProcGen benchmark (Cobbe at al. 2020 https://arxiv.org/abs/1912.01588) and TextWorld (Cote et al. 2018 https://arxiv.org/abs/1806.11532), and the NetHack Learning Environment (Kuttler et al. https://arxiv.org/abs/2006.13760) are missing, being frameworks for looking RL in procedurally generated settings (and also good candidates for future environments to try RAPID in!!).

Regardless, this paper provides a valuable contribution is studying exploration in procedurally generated environments - with the core idea of rewarding historically good behavior being one well proven in the past. The motivation is clear and the experiments well designed - proving that there are indeed environments where this idea works on, within the limits of the assumptions made. I would see this work accepted.

---

> ### Author Response · Authors · 2020-11-17
> **Thank you very much for the encouragement!**
>
> Thank you very much for the encouragement! Please find some clarification as follows.
>
> **This method rewards good exploration behavior historically and as such depends on the ability of the agent to discover such trajectories in the first place.**
>
> We believe that this is the fundamental reason why it is challenging to explore in a model-free setting. Since we have no knowledge of the environment, we can not easily discover novel trajectories. What we can do is to analyze the exploration behaviors historically and adjust the exploration strategy accordingly. RAPID approaches this problem by reproducing the good exploration behaviors historically so that it will be more likely to discover novel episodes.
>
> **High extrinsic reward implies good exploration behavior. Both of these assumptions likely do not hold in environments with sparse/ill-placed/deceptive rewards, challenges that environments such as Montezuma's Revenge/Net Hack/Interactive Fiction games are notorious for. In such environments, it is not only difficult to discover good exploration behaviors but also the overall extrinsic rewards themselves often lead to dead ends and globally suboptimal trajectories. The global score, being effectively count-based, alleviates some of this, but it is likely that this form of intrinsic that is based on extrinsic reward will likely fail in such scenarios. It would strengthen the paper to see positive results in such an environment.**
>
> We agree that “High extrinsic reward implies good exploration behavior” may not hold in other environments. We believe the hyperparameter $w_0$ exactly addresses this issue. i.e., we could set $w_0$ to zero if the extrinsic reward turns out to be not helpful. We observe that there are indeed situations when extrinsic rewards help. For example, in KeyCorridor-S4-R3 and MiniWorld-Maze (Figure 20), the performance drops substantially if removing extrinsic rewards. In the updated version, we add a “Limitations and Discussions” section to discuss this aspect. We will test RAPID in these environments in our future work.
>
> **Minor: A few of citations to the ProcGen benchmark, TextWorld, NetHack Learning Environment**
>
> Thank you for pointing this out! All these citations have been added in the introduction as well as the “Limitations and Discussions” section.

---

> > ### Comment · AnonReviewer2 · 2020-11-23
> > **Response to Rebuttal**
> >
> > Thank you for these clarifications, especially regarding the ability of agents to find good exploration behaviors. I agree that this is a fundamental issue of modern tabula rasa model-free RL.
> >
> > Overall, I will maintain my score as an accept.

---

### Official Review · AnonReviewer1 · 2020-10-29
**Simple and effective approach, with strong empirical evaluation**

**Rating:** 7
**Confidence:** 3

**Review:**

Summary:
This paper tackles the problem of improving exploration in deep RL for procedurally-generated environments, where state-of-the-art exploration techniques typically fail. In the proposed approach, called RAPID, each agent-generated episode is evaluated with respect to its local exploration score (for the given episode), global exploration score (across all previous episodes), and extrinsic reward obtained. Episodes with high scores are stored in a replay buffer, and a policy is trained via behavioral cloning on batches of state-action pairs from this buffer. This policy is also used to produce the agent-generated episodes.

Recommendation:
The approach is simple and intuitive, and empirically improves exploration in procedurally-generated environments. Training on procedurally-generated environments is becoming more common and useful, for instance in domain randomization for sim-to-real transfer in robotics, so this approach would be relevant for the ICLR audience. I have some concerns and questions (detailed below), but overall I recommend acceptance.

Pros:
* The approach itself is simple and easy to implement.
* The paper is clearly writtten and well-motivated.
* The empirical evaluation is thorough: RAPID is compared against a suite of state-of-the-art baselines for exploration bonuses, exploration in procedurally-generated environments, and a self-imitation approach; there are also ablation studies and hyperparameter sensitivity studies. The ablation study comparing behavioral cloning versus exploration bonuses (where the per-episode exploration scores are given as part of the reward to the agent) is particularly interesting, given that existing approaches typically rely on the latter.
* The empirical evaluation is on a variety of domains, including both discrete-action and continuous control tasks.

Cons:
* I would like to see an empirical comparison against Never Give Up (NGU; Badia et al. 2020), which also uses episodic novelty and global novelty to guide exploration. Although NGU uses the same environment for training and testing, because it takes into account how controllable a state is, it wouldn't suffer from the limitation highlighted in Figure 1 (where a state is randomly generated, regardless of the agent's action), and may do well in procedurally-generated environments.
* It seems strange to me that single state-action pairs are stored in the replay buffer, rather than keeping all state-action pairs from an entire episode together. It's possible that a particular state-action pair may only be good in terms of exploration, _in the context of_ the rest of the agent's trajectory in that episode.
* The continuous control experiments for MuJoCo locomotion tasks are contrived, since the extrinsic reward for forward progress is summed and given at the end of the episode, which just doesn't make sense for locomotion tasks, and unnecessarily hampers the baseline approaches. Using a goal-reaching continuous control task would be more relevant: e.g., Swimmer in a procedurally-generated maze.
* I would like to see a discussion of limitations and failure cases for RAPID.
* There are minor grammatical errors and typos throughout the paper.

Questions:
* Can the local score and global score be applied to observations (e.g., image observations) directly, instead of states?
* Are the scores associated with state-action pairs in the replay buffer taken into account during sampling of batches, or ignored?
* How were the weights chosen for the total score, that is a weighted sum of the extrinsic reward, local score, and global score? The weight for the global score is very small compared to the others (i.e., 0.001 versus 0.1 and 1). Why is this the case? What happens when this weight is higher (e.g., is there some type of suboptimal behavior that emerges)?
* Why is it necessary to anneal the imitation learning to zero for some environments, but not others?
* What is the state space for each of the tasks, that's used to compute the local and global scores? It would be useful to have a table in the Appendix that summarizes this.

---

> ### Author Response · Authors · 2020-11-17
> **Thank you for the constructive suggestions and comments!**
>
> Thank you for the constructive suggestions and comments! Please see the response and clarification as follows.
>
> **I would like to see an empirical comparison against Never Give Up (NGU; Badia et al. 2020), which also uses episodic novelty and global novelty to guide exploration. Although NGU uses the same environment for training and testing, because it takes into account how controllable a state is, it wouldn't suffer from the limitation highlighted in Figure 1 (where a state is randomly generated, regardless of the agent's action), and may do well in procedurally-generated environments**
>
> We agree that the episodic novelty in NGU may benefit the exploration in procedurally-generated environments. However, we couldn’t find an open-source implementation of NGU (please let us know if there is one). While the idea of NGU is simple, correctly implementing NGU seems non-trivial since there are lots of details in the implementation. Thus, we are not able to provide a fair comparison with NGU given the time constraint. Alternatively, in the updated version, we discuss NGU and some other episodic memory methods in the “Episodic Memory” of the Related Work section. Compared with NGU that incorporates local and global novelty with intrinsic rewards, RAPID uses a ranking mechanism to make use of the historical episode-level good exploration data. As a result, some good experiences could be used multiple times, which may contribute to the sample efficiency of RAPID. An interesting question is: “what is the relationship between the ranking strategy and the intrinsic rewards, and whether and how they can be combined?”. We do plan to investigate this question as one of our future work.
>
> **It seems strange to me that single state-action pairs are stored in the replay buffer, rather than keeping all state-action pairs from an entire episode together. It's possible that a particular state-action pair may only be good in terms of exploration, in the context of the rest of the agent's trajectory in that episode.**
>
> Thank you for pointing this out. The reason we use state-action pairs is for the ease of implementation. While the idea is to select the whole episode, the behavior cloning actually samples a batch of state-action pairs instead of episodes. Thus, implementing the buffer as state-action pairs is straightforward and natural. To address your concern, we implement a variant that forces the buffer to keep the entire episode by exceptionally allowing the episode at the end of the buffer to exceed the buffer size. We show the result on MultiRoom-N7-S8 in Appendix H. We do not observe a clear difference. Thus, we believe our implementation is a reasonable approximation of storing the entire episode.
>
> **The continuous control experiments for MuJoCo locomotion tasks are contrived, since the extrinsic reward for forward progress is summed and given at the end of the episode, which just doesn't make sense for locomotion tasks, and unnecessarily hampers the baseline approaches. Using a goal-reaching continuous control task would be more relevant: e.g., Swimmer in a procedurally-generated maze.**
>
> We chose these MuJoCo tasks because they are benchmarks in continuous control. Delaying the reward to increase the difficulties is also adopted in previous work, such as [1] [2]. While we agree that it may not make sense in real-world locomotion tasks, we believe we can observe some insights of how the algorithm handles sparse rewards in this setting. We think Swimmer in a procedurally-generated maze is a great idea to test the algorithm, and we will implement and try this environment in our future work. In the updated version, we have conducted an additional experiment to make Swimmer procedurally-generated by inserting random physical properties in each new episode (Appendix I).
>
> [1] Zheng, Zeyu, Junhyuk Oh, and Satinder Singh. "On learning intrinsic rewards for policy gradient methods." NeurIPS, 2018.
>
> [2] Oh, J., Guo, Y., Singh, S., & Lee, H. (2018). Self-imitation learning. ICML, 2018
>
>
> **I would like to see a discussion of limitations and failure cases for RAPID.**
>
> In the updated version, we have added a section “Limitations and Discussions” to highlight and discuss three limitations about scoring methods, ranking buffer, and the behavior cloning, as well as the potential future directions.
>
> **There are minor grammatical errors and typos throughout the paper.**
>
> Thanks for the suggestions. We have proofread again and fixed some typos in the updated version.

---

> > ### Author Response · Authors · 2020-11-17
> > **cont.**
> >
> > **Can the local score and global score be applied to observations (e.g., image observations) directly, instead of states?**
> >
> > To test whether the local score can be applied in images, we have tried the MiniWorld Maze environment, where the observation space is 60*80 images. Since counting the states is not applicable in continuous state space, we can calculate the variance of the images in an episode (see Eq. 2 in the updated version). While we directly calculate the scores on the raw images in our experiments, we believe a better solution could be learning the state embeddings and then calculating the scores on the embeddings, which we will study in future work. For the global score, we have not adapted it to images yet. A possible way is to use pseudo-count. In the updated version, we have added the discussions of the images in the “Limitations and Discussions”.
> >
> > **Are the scores associated with state-action pairs in the replay buffer taken into account during sampling of batches, or ignored?**
> >
> > They are ignored. We use uniform sampling.
> >
> > **How were the weights chosen for the total score, that is a weighted sum of the extrinsic reward, local score, and global score? The weight for the global score is very small compared to the others (i.e., 0.001 versus 0.1 and 1). Why is this the case? What happens when this weight is higher (e.g., is there some type of suboptimal behavior that emerges)?**
> >
> > We fixed $w_0=1$ and grid-searched $w_1$ and $w_2$ from 9 values on MiniGrid-N7-S8. Then we directly used these hyperparameters on other MiniGrid tasks. In the updated version, we have provided sensitivity analysis and discussions in Figure 8, Appendix A.
> >
> > **Why is it necessary to anneal the imitation learning to zero for some environments, but not others?**
> >
> > We find that sometimes annealing can improve the sample efficiency. Sometimes annealing would degrade the performance. We speculate that it depends on whether exploration or exploitation is more helpful. In the updated version, we provide a detailed discussion of the possible reasons in Appendix J.
> >
> > **What is the state space for each of the tasks, that's used to compute the local and global scores? It would be useful to have a table in the Appendix that summarizes this.**
> >
> > Thank you for the comment. It is a great idea. In the updated version, the table is provided in Table 3, Appendix A.

---

> > > ### Comment · AnonReviewer1 · 2020-11-18
> > > **Response to Authors**
> > >
> > > Thank you for the detailed answers to my questions, and for including additional experiments and explanations in the paper. I appreciate the additional "Limitations and Discussions" section as well. I understand and agree that it's difficult to include NGU as a baseline, given that there's no open-source implementation available.
> > >
> > > I'm happy to see the paper accepted in its current state.

---

### Author Response · Authors · 2020-11-17
**Summary of the revisions**

We thank all the reviewers for the feedback and comments. The improvements in the updated paper are summarized below.

**Experiments:**
1. Comparison of keeping all the state-action pairs of an episode and not keeping all the state-action pairs to address the concern of Reviewer 1 (Figure 25, Appendix H).
2. Additional results on two procedurally-generated swimmer environments to address the concerns of Reviewer 1 and Reviewer 4 (Figure 26, Appendix I). These environments have been merged to our codebase and will be open-sourced for reproducibility.
3. Sensitivity analysis of the weights of the three scores to address the concerns of Reviewer 1 and Reviewer 4 (Figure 8, Appendix A).
4. Comparison between using annealing and not using annealing to answer the question of Reviewer 1 (Figure 27, Appendix J).
5. Ablation results of removing ranking according to the suggestions provided by reviewer 4 (Table 1).

**Writing:**
1. Formally defined the local score in continuous state space to clarify the confusion of Reviewer 1, Reviewer 3, Reviewer 4.
2. Added “Episodic Memory” to Related Work according to the suggestions of Reviewer 1.
3. Added the “Limitations and Discussion” Section to address the concerns of all the reviewers.
4. Summarized how the local scores are calculated in each environment to clarify the confusion of Reviewer 1, Reviewer 3, and Reviewer 4 (Table 3, Appendix A).
5. Described the selection of weights in Appendix A to address the concerns of Reviewer 1 and Reviewer 4.
6. Added more discussions of the roles of global and local scores in the ablation study to clarify the concern of Reviewer 3 (RQ2, Section 3.3).

**Minor:**
1. Added citations to some procedurally-generated environments, suggested by Reviewer 2.
2. Explicitly clarified that the MuJoCo tasks are singleton and added the procedurally-generated Swimmer in Section 3.4 to address the concern of Reviewer 4.
3. Fixed a few typos, including the ones posted by Reviewer 3.

---

### Author Response · Authors · 2020-11-25
**Summary of our contributions**

We sincerely thank all the reviewers for the support and the insightful comments to help improve the paper. As a closing remark, we highlight the contributions of our paper as follows.

1. We have explored a new exploration strategy at the episode-level. Unlike the previous studies that mainly focus on state-level intrinsic rewards, we study how we can quantify the episode-level exploration behaviors. We show that the episodic score may better generalize in procedurally-generated environments since it focuses more on the overall exploration behaviors instead of unimportant details in specific states.

2. To validate this idea, we have proposed a simple yet effective approach named RAPID. First, we quantify episodic exploration behaviors from three aspects: a local score that provides a per-episode view and is generalization-aware, a global score that provides a long-term view to capture historical information, and the extrinsic reward that utilizes the environmental feedback. Second, we propose to use a ranking buffer to effectively exploit the good episode-level exploration behaviors. We demonstrate that RAPID delivers strong results on MiniGrid benchmarks and is also effective in environments with continuous state/action spaces, such as the 3D Maze in MiniWorld and some MuJoCo tasks.

3. We will open-source our code to facilitate future studies in episode-level exploration and research in procedurally-generated environments. While our results have shown promise in the episode-level exploration bonus, we made many simple choices in RAPID. In the updated paper, we have highlighted several limitations and future directions to enable our idea in more complicated environments.

---

### Decision · Program_Chairs · 2021-01-07
**Final Decision**

**Decision:**

Accept (Poster)

**Comment:**

In order to learn good exploratory behaviors in settings where agents encounter diverse environments, the authors propose an approach which involves learning from episodes that exhibit good episode-level exploratory behaviors.  The innovation is in the scoring and learning from these episode-level behaviors rather than trying to come up with shorter timescale proxies of exploration.  In making this concrete, the authors propose to score trajectories based effectively on state coverage within an episode (i.e. good exploration corresponds to good state coverage) as well as by scoring episodes relative to one another and giving preference to episodes that explore less often encountered states.  To learn, the core algorithm interleaves standard RL updates with behavioral cloning updates using the best episodes of data, thereby training the policy to both solve the task and explore well at the episode level.

A weakness is that the paper uses low-level state in grid worlds and there is some ambiguity in applying this to settings with continuous states.  The authors discuss general strategies for dealing with these limitations as potential future work.

The reviewers were positive about the clarity of the text and felt the core idea that was proposed was simple and effective.  The authors put in solid effort to address reviewer concerns.  The most salient remaining concern, which I share, is that there will be challenges in scaling this approach to more complex environments with continuous state/observation spaces.

Overall, this paper had a consensus "accept" rating (7,7,7,6), and I endorse this as my decision.